# SUBSPACE KERNEL LEARNING ON TENSOR SEQUENCES

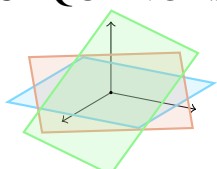

**Lei Wang**[1,2*]     **Xi Ding**[1*]     **Yongsheng Gao**[1†]     **Piotr Koniusz**[3,2,1†]
[1] Griffith University, [2] Data61♥CSIRO, [3] University of New South Wales
{l.wang4, x.ding, yongsheng.gao}@griffith.edu.au,
piotr.koniusz@unsw.edu.au

## ABSTRACT

Learning from structured multi-way data, represented as higher-order tensors, requires capturing complex interactions across tensor modes while remaining computationally efficient. We introduce Uncertainty-driven Kernel Tensor Learning (UKTL), a novel kernel framework for $M$-mode tensors that compares mode-wise subspaces derived from tensor unfoldings, enabling expressive and robust similarity measure. To handle large-scale tensor data, we propose a scalable Nyström kernel linearization with dynamically learned pivot tensors obtained via soft $k$-means clustering. A key innovation of UKTL is its uncertainty-aware subspace weighting, which adaptively down-weights unreliable mode components based on estimated confidence, improving robustness and interpretability in comparisons between input and pivot tensors. Our framework is fully end-to-end trainable and naturally incorporates both multi-way and multi-mode interactions through structured kernel compositions. Extensive evaluations on action recognition benchmarks (NTU-60, NTU-120, Kinetics-Skeleton) show that UKTL achieves state-of-the-art performance, superior generalization, and meaningful mode-wise insights. This work establishes a principled, scalable, and interpretable kernel learning paradigm for structured multi-way and multi-modal tensor sequences.

## 1 INTRODUCTION

The rapid growth of high-dimensional, multi-modal data, from video streams to biomedical signals, demands learning frameworks capable of capturing rich multi-way structures. Tensors, or multi-way arrays, naturally represent such data by preserving relationships across multiple modes such as space, time, and features (Kolda & Bader, 2009; Koniusz et al., 2013; Koniusz & Cherian, 2016; Koniusz et al., 2021; Wang et al., 2024b). Despite their expressiveness, tensors introduce unique challenges for machine learning. Traditional approaches often flatten tensors into vectors or matrices before applying standard techniques such as Principal Component Analysis (PCA). This simplification destroys the inherent structure of tensors, leading to models that

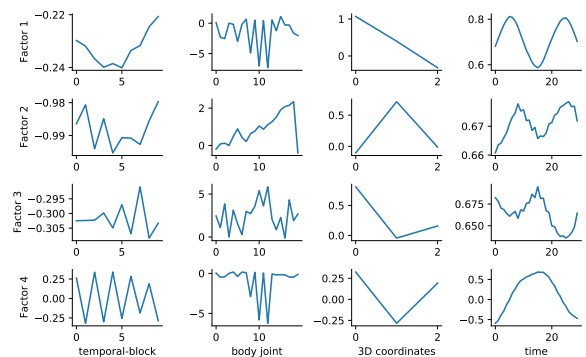

Figure 1: Mode-wise factor matrices from the Tucker decomposition for the action called *"draw x"*. Each row shows one latent factor (from 4 leading factors), and each column corresponds to one tensor mode: *temporal block*, *body joints*, *3D coordinates*, and *time*. Structured patterns reveal interpretable, mode-specific information which motivates our approach.

are inefficient or less expressive, especially when non-linear dependencies or sparse observations are present.

Kernel methods offer a powerful alternative by embedding data into high-dimensional Reproducing Kernel Hilbert Spaces (RKHS), enabling non-linear modeling through pairwise similarities without

---

*Equal contribution.     † Corresponding authors.

explicit feature mappings. However, most kernel methods rely on vectorized inputs, limiting their ability to directly use tensors. Furthermore, their computational cost grows rapidly with dataset size, impeding large-scale or real-time applications. On the other hand, tensor decomposition methods such as CP, Tucker, and Tensor-SVD provide effective low-rank representations that respect multi-way relations but are fundamentally linear and do not easily integrate with non-linear learning frameworks. Recent tensor subspace learning approaches seek to combine these advantages but often assume equal importance across tensor modes or resort to heuristic regularization.

In this work, we introduce a novel perspective: learning kernels over tensor subspaces with explicit modeling of uncertainty in each mode. Our key insight is that mode-wise unfolding of tensors reveals interpretable subspaces that can be robustly compared in RKHS using kernels (see Fig. 1). Building on this, we propose (i) a product subspace kernel that captures interactions across modes more effectively than direct tensor comparisons, (ii) a Nyström-based kernel linearization with dynamic pivot tensor selection via soft clustering for scalable training, and (iii) an uncertainty-aware regularization framework, grounded in likelihood maximization, that adaptively weights subspaces by their informativeness.

Our end-to-end framework learns non-linear, structured embeddings directly from tensor sequences, while maintaining computational efficiency. We validate our approach on challenging action recognition benchmarks, where it outperforms state-of-the-art graph convolutional, hypergraph, and transformer models, all with a simpler and more interpretable design. Our main **contributions** are:

i. We propose a principled kernel learning framework for tensor data that captures non-linear relationships through mode-wise subspace comparisons, avoiding destructive flattening or costly direct tensor matching.

ii. We introduce a Nyström kernel linearization with soft clustering to dynamically construct representative pivot tensors, enabling scalable training on large datasets.

iii. We develop an uncertainty-aware regularization that adaptively weights tensor subspaces based on their reliability using maximum likelihood estimation.

## 2 RELATED WORK

**Tensor decomposition and subspace learning.** Tensor decomposition methods have been widely adopted for learning low-dimensional representations of high-order data. Canonical models such as CANDECOMP/PARAFAC (CP) (Kruskal, 1977), Tucker decomposition (Lathauwer et al., 2000), and tensor singular value decomposition (t-SVD) (Kilmer et al., 2013) provide multilinear analogs to classical matrix factorizations such as PCA or SVD. These techniques exploit the inherent multi-mode structure of tensor data and are effective for compression, denoising, and subspace extraction. Building upon these foundations, approaches such as Multilinear PCA (MPCA) (Lu et al., 2008), Tensor Train PCA (TT-PCA) (Bengua et al., 2017), and higher-order embedding methods (Liu et al., 2010; Wang et al., 2018) adapt tensor decomposition to (semi-)supervised learning settings.

Despite their broad applicability, these methods are inherently linear and typically assume rigid structural constraints, such as orthogonality or fixed rank across modes (Lu et al., 2011). More critically, they often treat all tensor modes as equally informative, applying the same modeling assumptions regardless of the variation or relevance of each dimension (Wang & Koniusz, 2023; Zhang et al., 2025b; Dong et al., 2025). This uniform treatment limits results in real-world scenarios, where different modes (*e.g.*, spatial, temporal, semantic) may contribute unequally to the task at hand. Our approach diverges from traditional methods by introducing a non-linear, kernel-based formulation on mode-specific subspaces from tensor unfolding. Unlike standard tensor subspace techniques, we explicitly model mode-wise uncertainty, enabling adaptive, data-driven regularization based on subspace relevance.

**Kernel methods on tensor data.** Kernel methods offer powerful tools for capturing non-linear relationships in data through implicit mappings into high-dimensional RKHS. Early extensions of kernel PCA (Schölkopf et al., 1998) and support vector machines to tensor data often relied on flattening tensors into vectors or defining simple tensor-product kernels (Chen et al., 2022). More sophisticated formulations use structured kernels based on mode-wise decompositions or Kronecker-product designs (He et al., 2017), seeking to preserve multi-way dependencies within the kernel framework. While these methods successfully bring non-linearity into tensor-based modeling, they

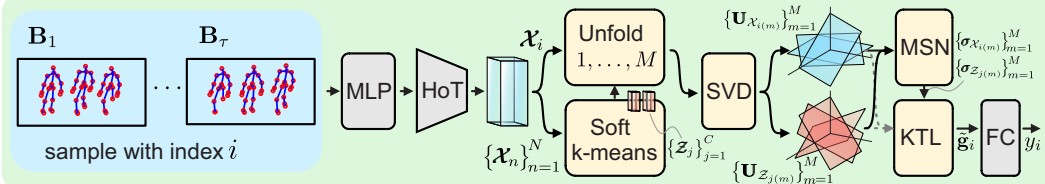

Figure 2: Overview of the proposed Uncertainty-driven Kernel Tensor Learning (UKTL) pipeline for action recognition. For brevity, we use skeletons as an example. Each skeleton sequence is divided into temporal blocks $\mathbf{B}_1, \ldots, \mathbf{B}_\tau$, embedded via an MLP, and processed by a Higher-order Transformer (HoT) to obtain feature tensor $\mathcal{X}_i$. These tensors undergo mode-$m$ matricization $(1, \ldots, M)$ and SVD to extract $M$ subspaces per sample. Soft $k$-means clustering yields $C$ Nyström pivots, each represented by also $M$ subspaces. A Multi-mode SigmaNet (MSN) estimates uncertainty vectors over all subspaces, which are used to regularize kernel computations. The Nyström-approximated KTL maps inputs to compact, uncertainty-aware representations, $\tilde{\mathbf{g}}_i$ for final classification. The entire model is trained end-to-end.

face significant challenges in practice. The most notable limitations include high computational cost due to dense kernel matrices, poor scalability to large datasets, and reliance on handcrafted or static kernel definitions. Moreover, they typically lack mechanisms for adapting kernel behavior based on the heterogeneous informativeness of different modes, a gap that is particularly critical for tasks involving dynamic or structured input data like tensor sequences.

Our method addresses these limitations through a product subspace kernel that compares tensors via their mode-wise unfolded bases. This structured formulation enables efficient, interpretable comparisons while preserving the multi-way organization of the input. Furthermore, our kernel is not predefined, but learned end-to-end using a Nyström approximation scheme with dynamically selected pivot tensors, allowing both scalability and adaptability to the underlying data distribution.

**Scalable kernel approximation.** The expressive power of kernel methods often comes at the cost of computational inefficiency, as computing and storing kernel matrices scales quadratically with the number of samples. To overcome this limitation, several approximation strategies have been proposed. The Nyström method (Williams & Seeger, 2000) approximates the full kernel matrix using a subset of representative samples, while random Fourier features (Rahimi & Recht, 2007) and convolution-based features (Koniusz et al., 2016) approximate shift-invariant kernels via explicit feature maps. Structured low-rank approximations (Gittens & Mahoney, 2016) further aim to reduce complexity by exploiting algebraic structure in the data.

Although effective in general settings, most existing kernel approximation techniques are designed for vector or matrix data, and often rely on static, randomly selected dictionary elements. These approximations are typically fixed prior to training and disconnected from the learning process, limiting their ability to adapt to task-specific data distributions. We introduce a Nyström-based approximation strategy tailored for tensor sequences, where pivots are constructed dynamically via soft $k$-means clustering. This approach allows the pivot set to evolve during training, ensuring that the kernel approximation remains relevant and responsive to the structure of both the sample tensors and the learned subspaces. Importantly, such an approximation mechanism is fully differentiable, enabling seamless integration into our end-to-end learning pipeline.

**Uncertainty modeling in representation learning.** Modeling uncertainty in neural representations enjoys growing attention, particularly in the context of Bayesian deep learning. Techniques such as variational inference (Blundell et al., 2015) and Monte Carlo dropout (Gal & Ghahramani, 2016) aim to quantify epistemic or aleatoric uncertainty in network predictions or parameter estimates. However, most existing methods focus on scalar outputs, latent features or rescaling distance (Wang & Koniusz, 2022) and do not extend uncertainty modeling to mode-related subspaces underlying tensor decompositions. In the domain of tensor learning, uncertainty is rarely modeled explicitly. Standard approaches apply uniform regularization across all tensor modes, implicitly assuming that each contributes equally to model performance. This assumption is problematic in applications where the signal strength, noise level, or semantic relevance differs substantially across modes. To address this gap, we propose a maximum likelihood-based framework for learning the relative uncertainty of mode-specific subspaces. This yields interpretable, data-driven regularization terms that

reflect the variability and informativeness of each subspace. By incorporating uncertainty directly into the learning objective, our method enhances both robustness and generalization, particularly in cases where certain modes are more prone to noise or contain less discriminative information.

# 3 PRELIMINARIES

**Notations.** We follow standard notational conventions in multilinear algebra (Vasilescu & Terzopoulos, 2003). Scalars are denoted by lowercase letters $(x)$, vectors by bold lowercase $(\mathbf{x})$, matrices by bold uppercase $(\mathbf{X})$, and higher-order tensors by calligraphic symbols $(\mathcal{X})$. The set $\mathcal{I}_N$ denotes the integer set $\{1, 2, \ldots, N\}$. An $M$-th order tensor $\mathcal{A} \in \mathbb{R}^{I_1 \times I_2 \times \cdots \times I_M}$ has elements indexed by $a_{i_1 \cdots i_M}$. The mode-$m$ unfolding (or matricization) of $\mathcal{A}$ is denoted $\mathcal{A}_{(m)} \in \mathbb{R}^{I_m \times \prod_{j \neq m} I_j}$, where each column corresponds to a mode-$m$ fiber (obtained by fixing all indices except $i_m$). The Frobenius norm of a tensor is given by $\|\mathcal{A}\|_F = \sqrt{\sum_{i_1, \ldots, i_M} a_{i_1 \cdots i_M}^2}$. For more details, see (Huckle, 2019).

**Higher-order Transformer (HoT) layers.** Let the HoT layer (Kim et al., 2021) be $f_{m \to n}$ : $\mathbb{R}^{J^m \times d} \to \mathbb{R}^{J^n \times d}$ with two sub-layers: (i) a higher-order self-attention $a_{m \to n} : \mathbb{R}^{J^m \times d} \to \mathbb{R}^{J^n \times d}$ and (ii) a feed-forward $\text{MLP}_{n \to n} : \mathbb{R}^{J^n \times d} \to \mathbb{R}^{J^n \times d}$. Moreover, let indexing vectors $\boldsymbol{i} \in \mathcal{I}_J^m \equiv \mathcal{I}_J \times \mathcal{I}_J \times \cdots \times \mathcal{I}_J$ ($m$ modes) and $\boldsymbol{j} \in \mathcal{I}_J^n \equiv \mathcal{I}_J \times \mathcal{I}_J \times \cdots \times \mathcal{I}_J$ ($n$ modes). For the input tensor $\mathbf{X} \in \mathbb{R}^{J^m \times d}$ with hyper-edges of order $m$, a HoT layer evaluates:

$$a_{m \to n}(\mathbf{X})_{\boldsymbol{j}} = \sum_{h=1}^{H} \sum_{\mu} \sum_{\boldsymbol{i}} \boldsymbol{\alpha}_{\boldsymbol{i}, \boldsymbol{j}}^{h, \mu} \mathbf{X}_{\boldsymbol{i}} \mathbf{W}_{h, \mu}^{V} \mathbf{W}_{h, \mu}^{O} \tag{1}$$

$$\text{MLP}_{n \to n}(a_{m \to n}(\mathbf{X})) = \text{L}_{n \to n}^{2}(\text{ReLU}(\text{L}_{n \to n}^{1}(a_{m \to n}(\mathbf{X})))), \tag{2}$$

$$f_{m \to n}(\mathbf{X}) = a_{m \to n}(\mathbf{X}) + \text{MLP}_{n \to n}(a_{m \to n}(\mathbf{X})), \tag{3}$$

where $\boldsymbol{\alpha}^{h, \mu} \in \mathbb{R}^{J^{m+n}}$ is the so-called attention coefficient tensor with $H$ heads, and $\boldsymbol{\alpha}_{\mathbf{i}, \mathbf{j}}^{h, \mu} \in \mathbb{R}^{J}$ is a vector, $\mathbf{W}_{h, \mu}^{V} \in \mathbb{R}^{d \times d_H}$ and $\mathbf{W}_{h, \mu}^{O} \in \mathbb{R}^{d_H \times d}$ are learnable parameters. Moreover, $\mu$ indexes over the so-called equivalence classes of order-$(m + n)$ in the same partition of nodes, $\text{L}_{n \to n}^{1} : \mathbb{R}^{J^n \times d} \to \mathbb{R}^{J^n \times d_F}$ and $\text{L}_{n \to n}^{2} : \mathbb{R}^{J^n \times d_F} \to \mathbb{R}^{J^n \times d}$ are equivariant linear layers and $d_F$ is the hidden dimension.

The HoT layers in our model operate directly on this hyper-edge tensor using the higher-order attention and feed-forward formulations provided in Eqs. 1 - 3. Specifically, the HoT attention term (Eq. 1) computes higher-order interactions among hyper-edges through the multi-head attention coefficients $\alpha_{\boldsymbol{i}, \boldsymbol{j}}^{h, \mu}$. The equivariant feed-forward is defined by $\text{L}_{n \to n}^{1}$ and $\text{L}_{n \to n}^{2}$ (Eq. 2); and the overall block applies a residual update followed by LayerNorm (Eq. 3). This two-layer HoT stack aggregates information across hyper-edges (and over temporal blocks, when applicable), producing the higher-order representation used by the subsequent tensor kernel module.

# 4 METHOD

In this paper, mode refers only to a tensor dimension (*e.g.*, spatial, temporal, hyper-edge) and not to sensor modality or multi-modal inputs. We propose an Uncertainty-driven Kernel Tensor Learning (UKTL) framework for action recognition (see Fig. 2). For simplicity, we illustrate our method using skeletons. Skeleton sequences are segmented into temporal blocks, embedded by an MLP, and encoded by a HoT to produce feature tensors. These tensors are unfolded along each mode and decomposed via SVD to extract subspace projections. Soft $k$-means clustering identifies representative Nyström pivot tensors, also projected into subspaces. A Multi-mode SigmaNet (MSN) estimates mode-wise uncertainty vectors, which regularize kernel computations, enhancing robustness. The Nyström approximation enables scalable, compact, and uncertainty-aware kernel embeddings, followed by classification. The entire pipeline is trained end-to-end for effective learning.

## 4.1 TENSOR REPRESENTATIONS OF SEQUENCES

**Definition of hyper-edges.** A hyper-edge in our model is defined as an *unordered triplet of joints*, *e.g.*, $\xi = (j_1, j_2, j_3)$, representing a three-way structural relation in the skeleton. We enumerate all valid joint triplets, so the total number of hyper-edges is $N_\xi = \binom{J}{3}$.

For skeletal action recognition, we consider two common forms: (i) raw tensors $\mathcal{X} \in \mathbb{R}^{d \times J \times T}$, where $d$ is the spatial dimensionality (*e.g.*, 2D/3D joint coordinates), $J$ is the number of body joints, and $T$ is the temporal length; and (ii) higher-order feature tensors $\mathcal{X} \in \mathbb{R}^{d \times N \times \tau}$, where $N$ is the number of (hyper-)edges and $\tau$ is the number of temporal blocks extracted from the sequence.

To keep the backbone lightweight while capturing high-order structure, we use a compact encoder composed of (a) a 3-layer MLP (FC-ReLU-FC-ReLU-Dropout-FC), and (b) a HoT (Kim et al., 2021). Each temporal block (with $T$ consecutive frames) is encoded into a $d \times J$ feature map by the MLP (the encoder produces per-joint embeddings $\boldsymbol{x}_{t,j} \in \mathbb{R}^d$ for each temporal block $t$). Let $\mathbf{X}_t \in \mathbb{R}^{d \times J}$ be the MLP output at temporal block $t \in \{1, \ldots, \tau\}$. A hyper-edge feature is then obtained by aggregating the three joint embeddings through the shared MLP. The HoT module then aggregates these (stacking hyper-edges across all triplets and temporal blocks forms the input tensor to HoT) to produce structured three-mode tensors encoding hyper-edge interactions.

To form the final input tensor for UKTL, we (i) permute the feature and hyper-edge modes; (ii) extract upper-triangular third-order hyper-edges (size $N_\xi = \binom{J}{3}$); and (iii) stack the tensors along the temporal axis. This results in a compact representation $\mathcal{X} \in \mathbb{R}^{d' \times N_\xi \times \tau}$, capturing rich multi-way spatial-temporal relationships.

**Mode-$m$ matricization & subspace representation.** Let $\mathcal{X} \in \mathbb{R}^{I_1 \times \cdots \times I_M}$ denote an $M$th-order tensor. The *mode-$m$ matricization* (also called unfolding) transforms $\mathcal{X}$ into a matrix $\boldsymbol{X}_{(m)} \in \mathbb{R}^{I_m \times \prod_{k \neq m} I_k}$ by rearranging its entries such that the $m$th mode becomes the row dimension and all other modes are flattened into the columns. The full set of matricizations is denoted as $\boldsymbol{X} = \{\boldsymbol{X}_{(1)}, \ldots, \boldsymbol{X}_{(M)}\}$. For our case, each skeleton sequence is encoded into a third-order tensor $\mathcal{X}_i \in \mathbb{R}^{d' \times N_\xi \times \tau}$, where $d'$ is the feature dimension, $N_\xi = \binom{J}{3}$ is the number of third-order hyper-edges, and $\tau$ is the number of temporal blocks. We apply mode-$m$ matricization to each tensor $\mathcal{X}_i$, resulting in: $\boldsymbol{X}_{i(1)} \in \mathbb{R}^{d' \times (N_\xi \tau)}$, $\boldsymbol{X}_{i(2)} \in \mathbb{R}^{N_\xi \times (d' \tau)}$, and $\boldsymbol{X}_{i(3)} \in \mathbb{R}^{\tau \times (d' N_\xi)}$.

Each matrix $\boldsymbol{X}_{i(m)}$ captures the structure of $\mathcal{X}_i$ along mode $m$ by aligning its corresponding slices as rows. These matrices serve as inputs to subspace-based kernel functions. To obtain a compact representation, we perform a Tucker decomposition (Kolda & Bader, 2009) on each tensor:

$$\mathcal{X}_i = \mathcal{C}_i \times_1 \boldsymbol{U}_{i(1)} \times_2 \boldsymbol{U}_{i(2)} \times_3 \boldsymbol{U}_{i(3)}, \tag{4}$$

where $\mathcal{C}_i$ is the core tensor, and $\boldsymbol{U}_{i(m)}$ contains the leading left singular vectors of $\boldsymbol{X}_{i(m)}$ (*i.e.*, the mode-$m$ subspace basis). These orthonormal matrices summarize the dominant variation in each mode and serve as the input to our product subspace kernel. Each satisfies:

$$\boldsymbol{X}_{i(m)} \boldsymbol{X}_{i(m)}^\top = \boldsymbol{U}_{i(m)} \boldsymbol{\Lambda}_{i(m)} \boldsymbol{U}_{i(m)}^\top, \tag{5}$$

where $\boldsymbol{\Lambda}_{i(m)}$ is a diagonal matrix of singular values. This decomposition ensures that our kernel compares tensor sequences in terms of their most informative multi-mode subspaces, leading to better generalization and robustness.

## 4.2 SUM-PRODUCT GRASSMANN KERNEL

While many kernel methods have been proposed for tensor data, few explicitly account for the inherent multi-linear structure of the tensor space. Signoretto et al. (2011) introduced a framework for RKHS of multilinear functions, allowing kernels to operate directly on tensors rather than on vectorized representations. This is achieved via bounded multilinear mappings $\psi : \mathcal{H}_1 \times \cdots \times \mathcal{H}_M \to \mathbb{R}$, where each $\mathcal{H}_m$ is an RKHS. These mappings form a Hilbert space of multilinear functions, effectively extending the kernel framework to infinite-dimensional tensor-valued functions. Given an RKHS embedding $\phi : \mathbb{R}^{I_1 \times \cdots \times I_M} \to \mathcal{H}$, the associated tensor kernel is defined as:

$$k(\mathcal{X}_i, \mathcal{X}_j) = \langle \phi(\mathcal{X}_i), \phi(\mathcal{X}_j) \rangle_\mathcal{H}. \tag{6}$$

To exploit mode-wise structural information, we adopt a product kernel formulation. Let $\mathcal{X}_{i(m)}$ and $\mathcal{X}_{j(m)}$ be the mode-$m$ matricizations of tensors $\mathcal{X}_i$ and $\mathcal{X}_j$. The product kernel is expressed as:

$$k(\mathcal{X}_i, \mathcal{X}_j) = \prod_{m=1}^{M} k(\mathcal{X}_{i(m)}, \mathcal{X}_{j(m)}), \tag{7}$$

where each factor kernel $k(\mathcal{X}_{i(m)}, \mathcal{X}_{j(m)})$ measures similarity between subspaces spanned by mode-$m$ unfoldings. Rather than comparing the raw unfoldings directly, which are high-dimensional and sensitive to noise, we project them onto low-dimensional subspaces via SVD. Specifically, for mode-$m$ matricization $\mathcal{X}_{(m)} \in \mathbb{R}^{I_m \times \bar{I}_m}$ (with $\bar{I}_m = \prod_{k \neq m} I_k$), we compute:

$$\mathcal{X}_{(m)} = \boldsymbol{U}_{\mathcal{X}_{(m)}} \boldsymbol{S}_{\mathcal{X}_{(m)}} \boldsymbol{V}_{\mathcal{X}_{(m)}}^\top, \tag{8}$$

where $\boldsymbol{U}_{\mathcal{X}_{(m)}} \in \mathbb{R}^{I_m \times p}$ contains the top-$p$ left singular vectors. This matrix defines a $p$-dimensional subspace, interpreted as a point on the Grassmann manifold $\mathrm{G}(p, I_m)$, the set of all $p$-dimensional linear subspaces in $\mathbb{R}^{I_m}$. We embed this point into the space of orthogonal projection matrices via:

$$\phi : \mathrm{G}(p, I_m) \to \mathbb{R}^{I_m \times I_m},$$
$$\mathrm{span}(\boldsymbol{U}_{\mathcal{X}_{(m)}}) \mapsto \boldsymbol{U}_{\mathcal{X}_{(m)}} \boldsymbol{U}_{\mathcal{X}_{(m)}}^\top. \tag{9}$$

This embedding is isometric with respect to the projection metric, enabling the use of Euclidean distances in the space of projection matrices. The projection kernel, which defines similarity on the Grassmann manifold, is given by:

$$k(\boldsymbol{U}_{\mathcal{X}_{i(m)}}, \boldsymbol{U}_{\mathcal{X}_{j(m)}}) = \|\boldsymbol{U}_{\mathcal{X}_{i(m)}}^\top \boldsymbol{U}_{\mathcal{X}_{j(m)}}\|_F^2, \tag{10}$$

and is known to be positive definite. Inspired by the Gaussian RBF kernel, we extend Eq. 10 to define a Grassmann-based factor kernel for each mode:

$$k(\mathcal{X}_{i(m)}, \mathcal{X}_{j(m)}) = \exp\left(-\frac{\|\boldsymbol{U}_{\mathcal{X}_{i(m)}} \boldsymbol{U}_{\mathcal{X}_{i(m)}}^\top - \boldsymbol{U}_{\mathcal{X}_{j(m)}} \boldsymbol{U}_{\mathcal{X}_{j(m)}}^\top\|_F^2}{2\sigma^2}\right). \tag{11}$$

Combining across all $M$ tensor modes, the final product kernel becomes:

$$k(\mathcal{X}_i, \mathcal{X}_j) = \prod_{m=1}^{M} \exp\left(-\frac{\|\boldsymbol{U}_{\mathcal{X}_{i(m)}} \boldsymbol{U}_{\mathcal{X}_{i(m)}}^\top - \boldsymbol{U}_{\mathcal{X}_{j(m)}} \boldsymbol{U}_{\mathcal{X}_{j(m)}}^\top\|_F^2}{2\sigma^2}\right). \tag{12}$$

This kernel is positive definite due to the closure properties of kernels (the product of valid kernels is valid). For flexibility, we also introduce a sum kernel:

$$k(\mathcal{X}_i, \mathcal{X}_j) = \sum_{m=1}^{M} \exp\left(-\frac{\|\boldsymbol{U}_{\mathcal{X}_{i(m)}} \boldsymbol{U}_{\mathcal{X}_{i(m)}}^\top - \boldsymbol{U}_{\mathcal{X}_{j(m)}} \boldsymbol{U}_{\mathcal{X}_{j(m)}}^\top\|_F^2}{2\sigma^2}\right). \tag{13}$$

**Kernelized Tensor Learning (KTL).** Finally, to balance global and local structure, we propose a sum-product kernel:

$$k(\mathcal{X}_i, \mathcal{X}_j) = \mu \sum_{m=1}^{M} k(\mathcal{X}_{i(m)}, \mathcal{X}_{j(m)}) + (1-\mu) \prod_{m=1}^{M} k(\mathcal{X}_{i(m)}, \mathcal{X}_{j(m)}), \tag{14}$$

where $\mu \in [0, 1]$ controls the relative contribution of the sum and product components.

> Representing tensors via mode-wise subspaces on Grassmann manifolds offers two key benefits: (i) computational efficiency, as subspaces are significantly lower-dimensional than raw unfoldings, and (ii) robustness to noise and occlusion, since the subspace can effectively approximate missing information. These properties make our Grassmann-based factor kernel particularly suitable for high-dimensional, structured, and often noisy data, *e.g.*, skeleton sequences.

### 4.3 UNCERTAINTY-DRIVEN SUBSPACE LEARNING

**Uncertainty-driven KTL (UKTL).** Different tensor modes may exhibit varying levels of uncertainty due to noise, missing data, or modality imbalance. To capture this, we model uncertainty individually across each mode rather than assuming a global, constant uncertainty as in prior works (Matthies, 2007; Kiureghian & Ditlevsen, 2009; Indrayan, c2008.; Hüllermeier & Waegeman, 2021; Kendall & Gal, 2017). Inspired by (Wang & Koniusz, 2022; 2025), we introduce a mode-wise uncertainty-aware mechanism called Multi-mode SigmaNet (MSN).

The MSN consists of $M$ branches, one per tensor mode. Each branch takes the projection matrix $\boldsymbol{U}_{\mathcal{X}_{i(m)}} \boldsymbol{U}_{\mathcal{X}_{i(m)}}^{\top}$ (*i.e.*, the mode-$m$ subspace of $\mathcal{X}_i$) and outputs a corresponding uncertainty vector $\boldsymbol{\sigma}_{\mathcal{X}_{i(m)}} \in \mathbb{R}^p$. Each branch is composed of an FC layer followed by a scaled sigmoid activation to ensure positivity and boundedness: $\boldsymbol{\sigma}_{\mathcal{X}_{i(m)}} = \text{ScaledSigmoid}(\text{FC}(\boldsymbol{U}_{\mathcal{X}_{i(m)}} \boldsymbol{U}_{\mathcal{X}_{i(m)}}^{\top}))$. To incorporate uncertainty into the kernel, we define:

$$\widetilde{\boldsymbol{U}}_{\mathcal{X}_{i(m)}} = \boldsymbol{U}_{\mathcal{X}_{i(m)}} / \sqrt{\boldsymbol{\sigma}_{\mathcal{X}_{i(m)}}}, \tag{15}$$

where the division is element-wise across the rows of $\boldsymbol{U}_{\mathcal{X}_{i(m)}}$, effectively down-weighting directions with higher uncertainty. We update our projection-based kernels (Eqs. 12 and 13) by replacing $\boldsymbol{U}_{\mathcal{X}_{i(m)}}$ with $\widetilde{\boldsymbol{U}}_{\mathcal{X}_{i(m)}}$. The resulting kernels are more robust to noisy or uncertain features in each tensor mode, and similarly for the sum-product kernel (Eq. 14).

## 4.4 NYSTRÖM KERNEL LINEARIZATION

**Pivot selection.** To reduce the computational complexity of kernel methods on tensorial data, we use a Nyström-based low-rank approximation of the kernel matrix. This allows us to obtain an explicit, finite-dimensional feature representation from the proposed tensor kernel. Given $N$ tensors $\{\mathcal{X}_i\}_{i=1}^N$, we first identify $C$ pivots $\{\mathcal{Z}_j\}_{j=1}^C$ through soft $k$-means clustering (Koniusz & Mikolajczyk, 2011), which is differentiable and suitable for end-to-end learning. Each pivot $\mathcal{Z}_j \in \mathbb{R}^{d' \times N_\xi \times \tau}$ approximates a local cluster center (*a.k.a.* local prototype) in the tensor space. We solve:

$$\min_{[\mathcal{Z}_1, \ldots, \mathcal{Z}_C]} \sum_{i=1}^N \left\| \mathcal{X}_i - \sum_{j=1}^C \mathcal{Z}_j \left[ \boldsymbol{\alpha}_i \right]_j \right\|_{\text{F}}^2, \tag{16}$$

where $\boldsymbol{\alpha}_i \in \mathbb{R}^C$ denotes $\mathcal{X}_i$'s soft assignment to the $C$ pivots.

**Nyström approximation.** Let $\mathbf{K}_{NC} \in \mathbb{R}^{N \times C}$ be the kernel matrix between the data and pivots, and $\mathbf{K}_{CC} \in \mathbb{R}^{C \times C}$ the kernel matrix among pivots:

$$[\mathbf{K}_{NC}]_{ij} = k(\mathcal{X}_i, \mathcal{Z}_j), \quad [\mathbf{K}_{CC}]_{ij} = k(\mathcal{Z}_i, \mathcal{Z}_j). \tag{17}$$

We stabilize inversion via eigendecomposition of $\mathbf{K}_{CC}$:

$$\mathbf{K}_{CC} = \boldsymbol{U} \boldsymbol{\Lambda} \boldsymbol{U}^{\top}, \tag{18}$$

and compute the inverse square root via:

$$\mathbf{P}^{-1} = \boldsymbol{U} \boldsymbol{\Lambda}^{-1/2} \boldsymbol{U}^{\top}. \tag{19}$$

The Nyström feature embedding is then given by:

$$\mathbf{G} = \mathbf{K}_{NC} \mathbf{P}^{-1}, \quad \widetilde{\mathbf{G}} = \mathbf{G} - \overline{\mathbf{G}}, \tag{20}$$

where $\overline{\mathbf{G}}$ is the mean across columns of $\mathbf{G}$ to ensure centering. The resulting $\widetilde{\mathbf{G}} \in \mathbb{R}^{N \times C}$ serves as the low-rank linearized kernel features.

**End-to-end model integration.** Our complete model stacks four modules: a tensor encoder (MLP + HoT), MSN for subspace uncertainty modeling, a sum-product Grassmann kernel with Nyström kernel linearization, and a final FC classifier. Formally, the model function is:

$$f(\mathcal{X}; \mathcal{P}) = f\big(\text{FC}(\text{MSN}(\text{HoT}(\text{MLP}(\mathcal{X}; \mathcal{P}_{\text{MLP}}); \mathcal{P}_{\text{HoT}}); \mathcal{P}_{\text{MSN}}); \mathcal{P}_{\text{FC}})\big), \tag{21}$$

where $\mathcal{P} = [\mathcal{P}_{\text{MLP}}, \mathcal{P}_{\text{HoT}}]$ are encoder parameters, $\mathcal{P}_{\text{MSN}}$ for uncertainty, and $\mathcal{P}_{\text{FC}}$ for classification.

> Our framework is modality-agnostic and supports any input representable as a structured tensor, including skeletons, RGB frames, depth maps, and other sensor streams, without architectural changes. Each modality is encoded using an uncertainty-aware tensor kernel that captures intra-modal correlations across spatial, temporal, and channel dimensions. Modality-specific tensor representations are projected into a shared kernel space and fused via learnable weighted summation, preserving modality structure while modeling inter-modal complementarities. The fused embedding is then processed by MLP+HoT for end-to-end multi-modal learning.

Table 1: Results on NTU-60, NTU-120, and Kinetics-Skeleton. UKTL outperforms graph, hypergraph, and transformer models by using uncertainty-aware tensor kernels. All tensor methods use the same MLP + HoT backbone for fair comparison.

| | Method | NTU-60 | | NTU-120 | | Kinetics-Skeleton | |
|---|---|---|---|---|---|---|---|
| | | X-Sub(%) | X-View(%) | X-Sub(%) | X-Setup(%) | Top-1(%) | Top-5(%) |
| **Graph-based** | TCN (Kim & Reiter, 2017) | - | - | - | - | 20.3 | 40.0 |
| | ST-GCN (Yan et al., 2018) | 81.5 | 88.3 | 70.7 | 73.2 | 30.7 | 52.8 |
| | AS-GCN (Li et al., 2019) | 86.8 | 94.2 | 78.3 | 79.8 | 34.8 | 56.5 |
| | 2S-AGCN (Shi et al., 2019) | 88.5 | 95.1 | 82.5 | 84.2 | 36.1 | 58.7 |
| | NAS-GCN (Peng et al., 2020) | 89.4 | 95.7 | - | - | 37.1 | 60.1 |
| | Shift-GCN (Cheng et al., 2020) | 90.7 | 96.5 | 85.9 | 87.6 | - | - |
| | MS-G3D (Liu et al., 2020) | 91.5 | 96.2 | 86.9 | 88.4 | 38.0 | 60.9 |
| | Sym-GNN (Li et al., 2022) | 90.1 | 96.4 | - | - | 37.2 | 58.1 |
| | SSL (Yan et al., 2023) | 92.8 | 96.5 | 84.8 | 85.7 | - | - |
| | CTR-GCN (Zhang et al., 2025a) | 92.6 | 96.7 | 89.6 | 91.0 | - | - |
| | FD-GCN (Huo et al., 2025) | 92.8 | 96.7 | 89.4 | 90.7 | - | - |
| | DSDC-GCN (Zhuang et al., 2025) | 93.0 | 97.1 | 89.9 | 90.6 | 38.6 | **63.4** |
| **Hypergraph-based** | Hyper-GNN (Hao et al., 2021) | 89.5 | 95.7 | - | - | 37.1 | 60.0 |
| | DHGCN (Wei et al., 2021) | 90.7 | 96.0 | 86.0 | 87.9 | 37.7 | 60.6 |
| | SD-HGCN (He et al., 2021) | 90.9 | 96.7 | 87.0 | 88.2 | 37.4 | 60.5 |
| | Selective-HCN (Zhu et al., 2022) | 90.8 | 96.6 | - | - | 38.0 | 61.1 |
| | Hyper-GCN (Zhou et al., 2024) | 91.4 | 95.5 | 87.0 | 88.7 | - | - |
| **Transformer-based** | ST-TR (Plizzari et al., 2021) | 90.3 | 96.3 | 85.1 | 87.1 | 38.0 | 60.5 |
| | STST (Zhang et al., 2021) | 91.9 | 96.8 | - | - | 38.3 | 61.2 |
| | MTT (Kong et al., 2022) | 90.8 | 96.7 | 86.1 | 87.6 | 37.9 | 61.3 |
| | 4s-GSTN (Jiang et al., 2022) | 91.3 | 96.6 | 86.4 | 88.7 | - | - |
| | MAMP (Mao et al., 2023) | 84.9 | 89.1 | 78.6 | 79.1 | - | - |
| | STJD-MP (Gunasekara et al., 2025) | 85.9 | 90.0 | 77.1 | 79.3 | - | - |
| **Tensor-based** | *Backbone* | 90.8 | 95.8 | 85.2 | 87.4 | 36.7 | 59.5 |
| | + KPCA (*baseline*) | 92.0 | 96.8 | 88.6 | 90.1 | 37.1 | 59.8 |
| | + TPCA (*baseline*) | 91.6 | 96.8 | 88.2 | 90.0 | 38.0 | 60.5 |
| | + KTL | 92.5 | 97.1 | 88.8 | 90.3 | 38.9 | 61.9 |
| | + UKTL | **93.1** | **97.3** | **90.0** | **91.4** | **39.2** | 62.3 |

**Training & inference.** The training loss combines classification and uncertainty regularization:

$$\ell^*(\mathcal{X}, \boldsymbol{y}; \mathcal{P}) = \sum_{i=1}^{N} \left[ \ell\left( f(\mathcal{X}_i; \mathcal{P}), y_i \right) + \beta \sum_{m=1}^{M} \sum_{k=1}^{p} \log\left( \frac{\sigma_{k, \mathcal{X}_{i(m)}} + 1}{\sum_j \sigma_{k, \mathcal{X}_{j(m)}} + 1} \right) \right], \qquad (22)$$

where $\ell(\cdot)$ is cross-entropy, $\beta$ weights the uncertainty regularization across modes. Moreover, $[\sigma_{1,\mathcal{X}}, \ldots, \sigma_{p,\mathcal{X}}]^\top = \boldsymbol{\sigma}_{\mathcal{X}}$.

At inference, the model (including learned pivots) remains fixed. Given $N'$ test sequences, the same pipeline is applied to generate predictions for sequence-level action classification. We now present our experiments, followed by detailed analysis and discussion.

## 5 EXPERIMENT

### 5.1 DATASETS AND SETUPS

**Datasets & protocols.** We evaluate our method on three large-scale benchmarks: (i) *NTU RGB+D (NTU-60)* (Shahroudy et al., 2016) contains 56,880 sequences across 60 action classes, featuring variable sequence lengths, high intra-class variability, and up to two subjects per clip, each with 25 joints. We follow two standard protocols: cross-subject (X-Sub) and cross-view (X-View). (ii) *NTU RGB+D 120 (NTU-120)* (Liu et al., 2019) extends NTU-60 to 120 classes and 114,480 samples, captured from 106 subjects and 155 viewpoints. We adopt the cross-subject (X-Sub) and cross-setup (X-Setup) evaluation protocols. We also evaluate other modalities (*e.g.*, RGB and depth) and their fusion on NTU-60 and NTU-120 using standard protocols. (iii) *Kinetics-Skeleton* is derived from the Kinetics dataset (Kay et al., 2017), with around 300,000 videos spanning 400 action categories. Skeletons are extracted using OpenPose (Cao et al., 2017) (18 joints per frame) as in ST-GCN (Yan et al., 2018). We use their released skeleton data and report Top-1 and Top-5 accuracy.

**Baselines.** We compare against several tensor-based baselines using a shared encoder (an MLP followed by a HoT block). Kernel PCA (KPCA) applies kernel functions to vectorized features, discarding tensor structure. Tensor PCA (TPCA) preserves multi-way structure via Tucker or HOSVD decomposition but remains inherently linear. Our proposed KTL introduces mode-wise kernel functions over tensor subspaces, enabling non-linear, structure-aware similarity. Uncertainty-driven KTL

(UKTL) further enhances this by modeling mode-specific uncertainty to adaptively weight subspace contributions. All models are trained under identical settings for fair comparison.

**Setups.** Experiments are implemented in PyTorch and trained with SGD (momentum 0.9, weight decay 0.0001, batch size 32) and an initial learning rate of 0.1. For NTU-60/120, the LR decays $\times 10$ at epochs 40 and 50, ending at 60; for Kinetics-Skeleton, decay occurs at 50 and 60, ending at 80. Models use two HoT layers for NTU-60/120 and four for Kinetics-Skeleton, with hidden dimension 16 and 4/8 attention heads, respectively. Videos are split into overlapping 30-frame blocks (stride 10). Hyperparameters (*e.g.*, $\mu$, $\beta$, Nyström pivots) are tuned via HyperOpt (Bergstra et al., 2015).

## 5.2 COMPARISONS WITH THE STATE OF THE ART

**Single-modality evaluation.** Table 1 shows that KTL and UKTL achieve strong performance across all three benchmarks. Graph-based methods like CTR-GCN and FD-GCN perform well, leveraging spatial-temporal skeleton modeling via graph convolutions. Hypergraph-based approaches further benefit from capturing higher-order joint relationships, while transformer-based models improve performance by modeling long-range dependencies with global attention.

Our tensor-based approaches, built on a consistent encoder backbone, systematically outperform these competing paradigms. KPCA improves over the backbone by introducing non-linear feature mappings but loses the inherent multi-way structure of the skeleton data. TPCA respects tensor modes and their multilinear structure but remains limited by its linear nature, showing improvements mainly on certain metrics. KTL advances these baselines by learning structured, non-linear kernel functions on tensor subspaces, effectively combining the advantages of non-linearity with tensor-mode awareness. This results in clear gains across all evaluation metrics, evidencing better feature discrimination and robustness. Crucially, UKTL, which incorporates uncertainty-driven regularization to adaptively weight different tensor modes based on their relevance and reliability, delivers the best overall performance. The consistent improvements of UKTL across all datasets and metrics, such as a 0.6% boost over KTL on NTU-60 X-Sub and 1.2% on NTU-120 X-Sub, highlight the effectiveness of modeling uncertainty in complex skeletal data representations.

> These results collectively underscore two key insights: (i) maintaining the tensor structure is vital for capturing the rich multi-dimensional correlations inherent in the data, *e.g.*, for skeletal action recognition; (ii) adapting model behavior through uncertainty estimation leads to more robust and discriminative representations, enhancing generalization in challenging scenarios.

**Multi-modal evaluation.** Table 2 presents the results. RGB and depth inputs exhibit spatial-temporal structures fundamentally different from skeletons. Nevertheless, our method achieves competitive RGB-only (83-86%) and depth-only (85-87%) performance. This confirms that the framework is not specialized to skeleton geometry and can effectively encode heterogeneous modalities once represented as tensors. Fusing Skeleton+RGB or Skeleton+Depth consistently yields significant improvements, reaching up to 95% on NTU-60 X-Sub. These gains indicate that kernel-based fusion successfully exploits complementary cues, *e.g.*, motion structure from skeletons, appearance from RGB, and geometry from depth, without modality-specific architectures. Even RGB+Depth fusion, without skeleton input, enjoys improvements over single-modality baselines due to effective inter-modal modeling.

Table 2: Evaluation of single- and multi-modal performance on NTU-60 and NTU-120. Skeleton alone provides a strong, competitive baseline, while RGB and Depth demonstrate the framework's flexibility across modalities. Fusing multiple modalities consistently improves accuracy, showing that UKTL effectively captures complementary information and inter-modal correlations.

| Modality | NTU-60 | | NTU-120 | |
|---|---|---|---|---|
| | X-Sub (%) | X-View (%) | X-Sub (%) | X-Setup (%) |
| Skeleton | 93.1 | 97.3 | 90.0 | 91.4 |
| RGB | 83.2 | 86.1 | 79.0 | 81.0 |
| Depth | 85.0 | 87.5 | 80.5 | 82.1 |
| Skeleton + RGB | 94.5 | 97.9 | 91.3 | 92.5 |
| Skeleton + Depth | 94.8 | 98.0 | 91.5 | 92.7 |
| RGB + Depth | 87.2 | 89.6 | 82.5 | 84.3 |
| Skeleton + RGB + Depth | **95.5** | **98.5** | **92.8** | **94.0** |

Combining Skeleton, RGB, and Depth achieves the best overall results (95.5/98.5 on NTU-60 and 92.8/94.0 on NTU-120), outperforming both uni-modal baselines and multi-stream models. All fusion is performed within a unified tensor-kernel formulation, without ad-hoc modality-specific

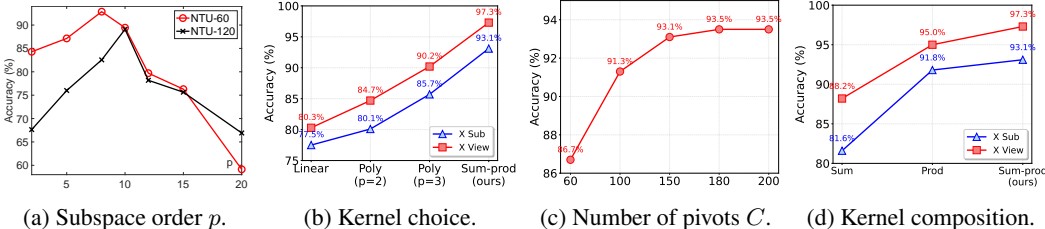

| (a) Subspace order $p$. | (b) Kernel choice. | (c) Number of pivots $C$. | (d) Kernel composition. |

Figure 3: Ablation study evaluating the effects of subspace order (on NTU-60/120), and of kernel choice, Nyström pivots, and kernel composition (on NTU-60) within the UKTL framework.

components. Although evaluated on human action datasets, the consistent gains across modalities and their combinations demonstrate that UKTL generalizes beyond skeleton-based inputs and naturally extends to other structured tensor modalities.

These results suggest that UKTL's gains arise from its ability to operate in a shared tensor-kernel space where modality-specific structure is preserved while alignment occurs implicitly through kernel interactions, rather than through explicit architectural coupling. This implicit alignment makes fusion resilient to modality imbalance and noise, allowing informative modalities to dominate when present without suppressing weaker ones. Moreover, the absence of modality-specific design choices indicates that performance improvements stem from the kernel formulation itself, highlighting UKTL as a scalable and principled approach for multi-modal learning in settings where modality availability, quality, or structure may vary.

## 5.3 ABLATION STUDY

Below, we conduct an ablation study on skeleton sequences from NTU-60 and NTU-120 to evaluate key hyperparameters and configurations. See Appendix for additional details and evaluations.

**Subspace order $p$.** Fig. 3a shows that increasing the subspace order $p$ consistently improves accuracy on NTU-60 and NTU-120 (X-Sub), with performance saturating beyond a dataset-specific threshold. This suggests that low-dimensional subspaces effectively capture the essential discriminative structure. Optimal values are $p = 8$ for NTU-60 and $p = 10$ for NTU-120.

**Kernel choice.** To evaluate the effectiveness of our kernel (Eq. 14), we compare it with standard linear and polynomial kernels. As shown in Fig. 3b, the linear kernel performs worst (77.5% on X-Sub), while polynomial variants offer moderate gains. In contrast, our sum-product kernel achieves 93.1% (X-Sub) and 97.3% (X-View), highlighting its superior ability to capture complex subspace interactions by jointly modeling additive and multiplicative relationships.

**Number of Nyström pivots.** We examine how UKTL's performance varies with the number of Nyström pivots used for kernel approximation. As shown in Fig. 3c, accuracy improves significantly up to 150 pivots, then stabilizes beyond 180, reflecting a balance between approximation quality and computational cost. To ensure optimal performance across datasets, we use HyperOpt (Bergstra et al., 2015) to automatically tune the pivot number during training.

**Kernel composition.** Below we compare sum-only, product-only, and combined (sum-product) kernels. As shown in Fig. 3d, the sum-kernel yields 81.6% (X-Sub), while the product-kernel achieves 91.8%. Their combination reaches the highest accuracy, 93.1% (X-Sub) and 97.3% (X-View), demonstrating that additive and multiplicative structures capture complementary information.

## 6 CONCLUSION

We introduced *Uncertainty-driven Kernel Tensor Learning (UKTL)*, a scalable and principled framework for structured multi-way and multi-modal data. UKTL combines mode-wise subspace kernels, dynamic Nyström linearization, and uncertainty-aware regularization to enable expressive, adaptive, and efficient tensor comparison without flattening or losing structure. Experiments on action recognition datasets show UKTL consistently surpasses state-of-the-art graph, hypergraph, and transformer models, while offering interpretable mode-wise insights. This establishes a powerful kernel-based approach for learning from structured tensor data.

ACKNOWLEDGMENTS

Xi Ding, a visiting scholar at the ARC Research Hub for Driving Farming Productivity and Disease Prevention, Griffith University, conducted this work under the supervision of Lei Wang. Lei Wang proposed the algorithm and developed the theoretical framework, while Xi Ding implemented the code and performed the experiments. We thank the anonymous reviewers for their invaluable insights and constructive feedback, which have contributed to improving our work.

This work was supported by the Australian Research Council (ARC) under Industrial Transformation Research Hub Grant IH180100002. This work was also supported by the CSIRO Allocation Scheme (Lead CI: Piotr Koniusz), the National Computational Merit Allocation Scheme 2025 (NCMAS 2025; Lead CI: Lei Wang) and the ANU Merit Allocation Scheme (ANUMAS 2025; Lead CI: Lei Wang), with computational resources provided by NCI Australia, an NCRIS-enabled capability supported by the Australian Government.

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

## A  DERIVATIVE OF UNCERTAINTY-DRIVEN KERNELIZED TENSOR LEARNING

For two $M$-order tensors $\mathcal{X}_i$ and $\mathcal{X}_j$, we design a kernel function that captures their multi-mode structure via uncertainty-aware subspace similarities. Specifically, we formulate two variants of the kernel, sum and product, based on mode-wise subspace comparisons. These are defined as follows.

**Uncertainty-driven sum kernel:**

$$
\begin{aligned}
k(\mathcal{X}_i, \mathcal{X}_j) &= \sum_{m=1}^{M} k(\mathcal{X}_{i(m)}, \mathcal{X}_{j(m)}) \\
&= \sum_{m=1}^{M} \exp\left(-\frac{\|\widetilde{\boldsymbol{U}}_{\mathcal{X}_{i(m)}}\widetilde{\boldsymbol{U}}_{\mathcal{X}_{i(m)}}^{\top} - \widetilde{\boldsymbol{U}}_{\mathcal{X}_{j(m)}}\widetilde{\boldsymbol{U}}_{\mathcal{X}_{j(m)}}^{\top}\|_{\mathrm{F}}^{2}}{2\sigma^2}\right)
\end{aligned}
\tag{23}
$$

**Uncertainty-driven product kernel:**

$$
\begin{aligned}
k(\mathcal{X}_i, \mathcal{X}_j) &= \prod_{m=1}^{M} k(\mathcal{X}_{i(m)}, \mathcal{X}_{j(m)}) \\
&= \prod_{m=1}^{M} \exp\left(-\frac{\|\widetilde{\boldsymbol{U}}_{\mathcal{X}_{i(m)}}\widetilde{\boldsymbol{U}}_{\mathcal{X}_{i(m)}}^{\top} - \widetilde{\boldsymbol{U}}_{\mathcal{X}_{j(m)}}\widetilde{\boldsymbol{U}}_{\mathcal{X}_{j(m)}}^{\top}\|_{\mathrm{F}}^{2}}{2\sigma^2}\right)
\end{aligned}
\tag{24}
$$

Here, $k(\mathcal{X}_i, \mathcal{X}_j)$ denotes the overall tensor kernel, while $k(\mathcal{X}_{i(m)}, \mathcal{X}_{j(m)})$ is the $m$-th mode kernel based on mode-$m$ matricizations $\mathcal{X}_{i(m)}$ and $\mathcal{X}_{j(m)}$. The matrices $\widetilde{\boldsymbol{U}}_{\mathcal{X}_{i(m)}}$ and $\widetilde{\boldsymbol{U}}_{\mathcal{X}_{j(m)}}$ represent the uncertainty-weighted subspaces derived via singular value decomposition (SVD) and reweighted by uncertainty estimates.

**Uncertainty-driven sum-product kernel:** We combine the sum and product kernels to form a more expressive hybrid:

$$
\begin{aligned}
k(\mathcal{X}_i, \mathcal{X}_j) =\, &\mu \sum_{m=1}^{M} k(\mathcal{X}_{i(m)}, \mathcal{X}_{j(m)}) + (1-\mu)\prod_{m=1}^{M} k(\mathcal{X}_{i(m)}, \mathcal{X}_{j(m)}) \\
=\, &\mu \sum_{m=1}^{M} \exp\left(-\frac{\left\|\widetilde{\boldsymbol{U}}_{\mathcal{X}_{i(m)}}\widetilde{\boldsymbol{U}}_{\mathcal{X}_{i(m)}}^{\top} - \widetilde{\boldsymbol{U}}_{\mathcal{X}_{j(m)}}\widetilde{\boldsymbol{U}}_{\mathcal{X}_{j(m)}}^{\top}\right\|_{\mathrm{F}}^{2}}{2\sigma^2}\right) \\
&+ (1-\mu)\prod_{m=1}^{M} \exp\left(-\frac{\left\|\widetilde{\boldsymbol{U}}_{\mathcal{X}_{i(m)}}\widetilde{\boldsymbol{U}}_{\mathcal{X}_{i(m)}}^{\top} - \widetilde{\boldsymbol{U}}_{\mathcal{X}_{j(m)}}\widetilde{\boldsymbol{U}}_{\mathcal{X}_{j(m)}}^{\top}\right\|_{\mathrm{F}}^{2}}{2\sigma^2}\right) \\
=\, &\mu \sum_{m=1}^{M} \exp\left(-\frac{\left\|\frac{\boldsymbol{U}_{\mathcal{X}_{i(m)}}\boldsymbol{U}_{\mathcal{X}_{i(m)}}^{\top}}{\boldsymbol{\sigma}_{\mathcal{X}_{i(m)}}} - \frac{\boldsymbol{U}_{\mathcal{X}_{j(m)}}\boldsymbol{U}_{\mathcal{X}_{j(m)}}^{\top}}{\boldsymbol{\sigma}_{\mathcal{X}_{j(m)}}}\right\|_{\mathrm{F}}^{2}}{2\sigma^2}\right) \\
&+ (1-\mu)\prod_{m=1}^{M} \exp\left(-\frac{\left\|\frac{\boldsymbol{U}_{\mathcal{X}_{i(m)}}\boldsymbol{U}_{\mathcal{X}_{i(m)}}^{\top}}{\boldsymbol{\sigma}_{\mathcal{X}_{i(m)}}} - \frac{\boldsymbol{U}_{\mathcal{X}_{j(m)}}\boldsymbol{U}_{\mathcal{X}_{j(m)}}^{\top}}{\boldsymbol{\sigma}_{\mathcal{X}_{j(m)}}}\right\|_{\mathrm{F}}^{2}}{2\sigma^2}\right)
\end{aligned}
\tag{25}
$$

In this formulation: $\boldsymbol{U}_{\mathcal{X}_{i(m)}}$ and $\boldsymbol{U}_{\mathcal{X}_{j(m)}}$ are the unitary matrices obtained from SVD of mode-$m$ matricizations. $\boldsymbol{\sigma}_{\mathcal{X}_{i(m)}}$ and $\boldsymbol{\sigma}_{\mathcal{X}_{j(m)}}$ are uncertainty scores obtained via our proposed Multi-mode SigmaNet (MSN), which takes as input the projections $\boldsymbol{U}_{\mathcal{X}_{i(m)}}\boldsymbol{U}_{\mathcal{X}_{i(m)}}^{\top}$ and $\boldsymbol{U}_{\mathcal{X}_{j(m)}}\boldsymbol{U}_{\mathcal{X}_{j(m)}}^{\top}$, respectively. $\mu \in [0,1]$ controls the contribution of the sum versus product kernel: setting $\mu = 1$ results in

a pure sum kernel, while $\mu = 0$ yields a product kernel. $\sigma$ is the bandwidth of the RBF kernel, kept constant across all experiments.

> **Remarks.** The sum kernel captures independent mode-wise similarities, while the product kernel emphasizes joint agreement across all modes, leading to stricter similarity enforcement. By interpolating between them with $\mu$, the proposed sum-product kernel offers a flexible mechanism to balance robustness and expressivity. Moreover, the uncertainty weighting via MSN enables the model to emphasize more reliable subspace components, leading to more stable and discriminative tensor comparisons.

## B    MAXIMUM LIKELIHOOD INTERPRETATION OF MODE-WISE UNCERTAINTY

We provide a probabilistic justification for the uncertainty weighting mechanism in UKTL. In particular, we interpret the learned uncertainty vectors $\boldsymbol{\sigma}_{\mathcal{X}_{i(m)}}$ as estimates of mode-specific heteroscedastic noise variances, modeled via maximum likelihood.

**Probabilistic model.** Let $\boldsymbol{U}_{\mathcal{X}_{i(m)}} \in \mathbb{R}^{p \times r}$ denote the orthonormal basis of the mode-$m$ subspace for sample $\mathcal{X}_i$, obtained via truncated SVD. We assume each row $\boldsymbol{U}_{\mathcal{X}_{i(m)},k}$ is a noisy observation of a latent clean subspace vector $\boldsymbol{S}_{i(m),k} \in \mathbb{R}^r$ corrupted by Gaussian noise with component-specific variance:

$$\boldsymbol{U}_{\mathcal{X}_{i(m)},k} \sim \mathcal{N}\left(\boldsymbol{S}_{i(m),k}, \sigma_k \mathbf{I}\right), \quad k = 1, \ldots, p. \tag{26}$$

Here, $\boldsymbol{\sigma}_{\mathcal{X}_{i(m)}} = [\sigma_1, \ldots, \sigma_p]^\top$ captures the noise variances across the $p$ components of the mode-$m$ subspace.

**Maximum likelihood estimation.** Given the dataset, the total negative log-likelihood under this model is:

$$\mathcal{L}(\boldsymbol{\sigma}) = -\sum_{i,m} \log p\left(\boldsymbol{U}_{\mathcal{X}_{i(m)}} \mid \boldsymbol{S}_{i(m)}, \boldsymbol{\sigma}_{\mathcal{X}_{i(m)}}\right) \tag{27}$$

$$= \sum_{i,m} \sum_{k=1}^{p} \left(\frac{1}{\sigma_k} \|\boldsymbol{U}_{\mathcal{X}_{i(m)},k} - \boldsymbol{S}_{i(m),k}\|_2^2 + \log \sigma_k\right) + \text{const}, \tag{28}$$

which simplifies to a weighted Frobenius loss plus a log-barrier:

$$\min_{\boldsymbol{\sigma}} \sum_{i,m} \left\|\frac{\boldsymbol{U}_{\mathcal{X}_{i(m)}} - \boldsymbol{S}_{i(m)}}{\sqrt{\boldsymbol{\sigma}_{\mathcal{X}_{i(m)}}}}\right\|_F^2 + \lambda \sum_k \log \sigma_k. \tag{29}$$

The division is applied row-wise, and the $\log \sigma_k$ term acts as a regularizer to prevent variance explosion.

**Practical realization.** In practice, we do not have access to ground truth clean subspaces $\boldsymbol{S}_{i(m)}$. Instead, we approximate them using the observed $\boldsymbol{U}_{\mathcal{X}_{i(m)}}$ itself as input to a small network, MSN, which predicts the corresponding uncertainty vector $\boldsymbol{\sigma}_{\mathcal{X}_{i(m)}}$. To integrate this into the kernel computation, we define a scaled subspace basis:

$$\widetilde{\boldsymbol{U}}_{\mathcal{X}_{i(m)}} = \boldsymbol{U}_{\mathcal{X}_{i(m)}} / \sqrt{\boldsymbol{\sigma}_{\mathcal{X}_{i(m)}}}, \tag{30}$$

where division is performed row-wise. This transformation suppresses unreliable subspace directions by shrinking their contribution in the kernel, while preserving well-estimated ones.

**Uncertainty-aware kernel.** The transformed basis $\widetilde{\boldsymbol{U}}_{\mathcal{X}_{i(m)}}$ is then used in the kernel computation (see Eq. 14), yielding a kernel function that is robust to subspace noise.

> **Remarks.** This probabilistic formulation shows that our uncertainty-aware kernel arises naturally from a maximum likelihood perspective under heteroscedastic Gaussian noise. It enables UKTL to down-weight unstable or noisy subspace components and focus on discriminative structure, especially when mode-wise quality varies.

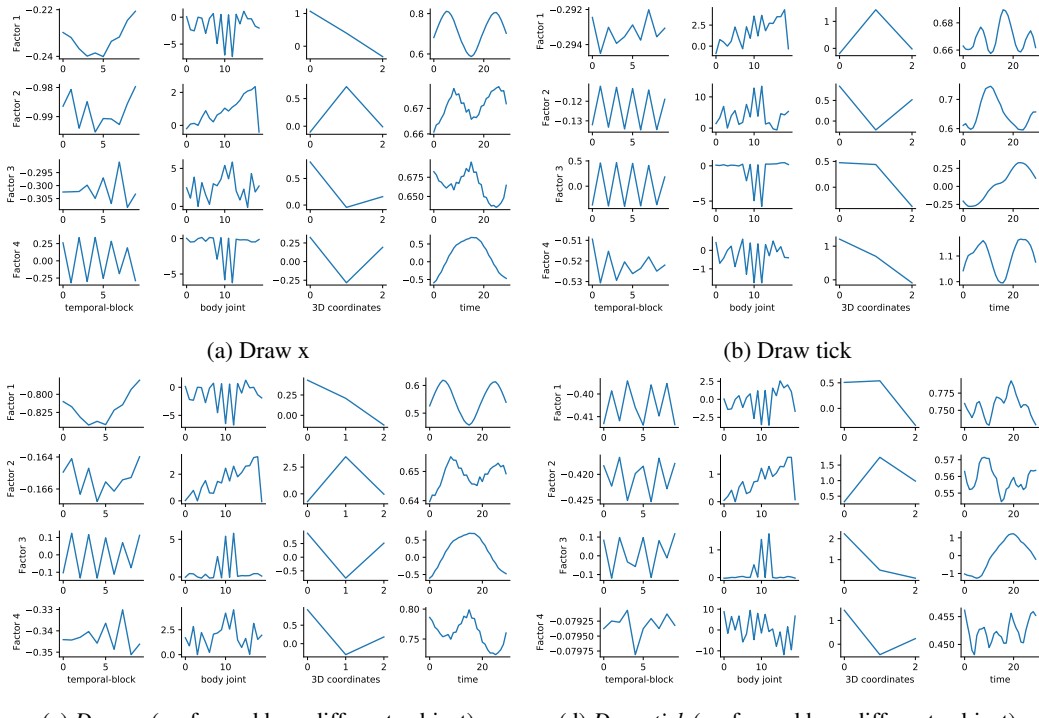

Figure 4: Visualization of Tucker decomposition in each mode of tensor representations for action *draw x* and action *draw tick*.

## C  SKELETON DATA PREPROCESSING

All skeleton sequences are processed to have a uniform temporal length of $T = 200$ across datasets. Specifically, (i) if a sequence has fewer than $T$ frames, we repeat it cyclically until it reaches length $T$; (ii) if it has $T$ or more frames, we uniformly sample $T$ frames. We then divide each sequence into temporal blocks of length $T^*$ using stride $S$, ensuring a consistent number of temporal blocks across all sequences for fair comparison.

Before feeding the skeleton sequences into the MLP and Higher-order Transformer (HoT) modules, we perform joint-wise normalization. First, each joint $\boldsymbol{v}_{f,i}$ in frame $f$ is centered relative to a reference joint (*e.g.*, the torso joint $\boldsymbol{v}_{f,c}$):

$$\boldsymbol{v}'_{f,i} = \boldsymbol{v}_{f,i} - \boldsymbol{v}_{f,c}, \tag{31}$$

where $i$ indexes the joint, and $c$ denotes the reference joint. Next, we normalize the coordinates of each joint into the range $[-1, 1]$ independently along each axis $j \in \{x, y, z\}$:

$$\hat{\boldsymbol{v}}_{f,i}[j] = \frac{\boldsymbol{v}'_{f,i}[j]}{\max_{f \in \{1, \dots, \tau\}, i \in \{1, \dots, J\}} \left| \boldsymbol{v}'_{f,i}[j] \right|}, \tag{32}$$

where $\tau$ is the number of frames and $J$ is the number of joints per frame.

For skeleton sequences involving multiple subjects, we handle each skeleton independently. Specifically, (i) each skeleton is normalized separately and processed individually through the MLP to learn its temporal dynamics; (ii) the resulting per-skeleton features are passed separately through the HoT block. Finally, we aggregate the HoT outputs from all subjects in a sequence using average pooling.

# D    MOTIVATION FOR KERNEL TENSOR LEARNING

To motivate our KTL framework, we analyze the basis components obtained from Tucker decomposition on raw tensor representations of two visually similar actions: *draw x* and *draw tick*.

Each skeleton sequence is represented as a tensor $\mathcal{X} \in \mathbb{R}^{\tau \times J \times 3 \times T^*}$, where $\tau$ denotes the number of temporal blocks, $J$ is the number of joints, and $T^*$ is the length of each temporal block. We apply Tucker decomposition to extract the basis matrices in each mode of the tensor.

As illustrated in Fig. 4, the decomposed bases for sequences of the same action label exhibit strong similarity across all modes. For example, the basis components for two different samples of *draw x* (Fig. 4a and Fig. 4c) are highly aligned, as are those for *draw tick* (Fig. 4b and Fig. 4d). In contrast, even though *draw x* and *draw tick* are semantically and visually similar actions in terms of 3D joint trajectories, their respective basis components reveal clear differences.

> This observation suggests that the mode-wise bases obtained through tensor decomposition encode discriminative and compact representations of action dynamics. Rather than comparing full high-dimensional tensors, we can use these subspace representations for more efficient and meaningful comparisons. This motivates our approach in KTL, where we model and compare tensor-mode subspaces using kernel methods to capture non-linear relationships among sequences.

# E    MANIFOLD GEOMETRY AND SUBSPACE LEARNING

**Symmetric Positive Definite (SPD) Manifolds.** The set of $d \times d$ symmetric positive definite (SPD) matrices, denoted $\mathrm{Sym}_d^+$, forms a Riemannian manifold, *i.e.*, a curved non-Euclidean space embedded in $\mathbb{R}^{d \times d}$. Riemannian geometry provides a more appropriate notion of dissimilarity for SPD matrices than the Euclidean metric.

A commonly used distance on $\mathrm{Sym}_d^+$ is the affine-invariant Riemannian metric:

$$d_{\mathrm{R}}(\mathbf{S}_1, \mathbf{S}_2) = \left\| \log \left( \mathbf{S}_1^{-1/2} \mathbf{S}_2 \mathbf{S}_1^{-1/2} \right) \right\|_F, \tag{33}$$

where $\mathbf{S}_1, \mathbf{S}_2 \in \mathrm{Sym}_d^+$, $\log(\cdot)$ is the matrix logarithm, and $\| \cdot \|_F$ denotes the Frobenius norm. This metric is invariant to affine transformations and aligns with the manifold's curvature.

In learning tasks, SPD matrices can be projected to lower-dimensional SPD spaces via bilinear maps of the form $\mathbf{W}^\top \mathbf{X} \mathbf{W}$, where $\mathbf{W} \in \mathbb{R}^{n \times m}$, $m < n$, ensuring that the projected matrix remains SPD (Harandi et al., 2018).

**Grassmann Manifolds.** The Grassmannian $\mathcal{G}(r, n)$ is the space of all $r$-dimensional linear subspaces of $\mathbb{R}^n$. A point on this manifold is represented by an orthonormal matrix $\mathbf{Y} \in \mathbb{R}^{n \times r}$ whose columns span the subspace. The geodesic distance between two subspaces $\mathrm{span}(\mathbf{Y})$ and $\mathrm{span}(\mathbf{Y}')$ is defined via their principal angles $\theta_k$ as:

$$d_{\mathrm{G}}(\mathbf{Y}, \mathbf{Y}') = \left( \sum_{k=1}^{r} \theta_k^2 \right)^{1/2}. \tag{34}$$

An alternative is to embed Grassmann points into the SPD cone using projection matrices $\mathbf{Y}\mathbf{Y}^\top$, enabling simpler kernel operations via the Frobenius norm:

$$\mathrm{dist}(\mathbf{Y}, \mathbf{Y}') = \|\mathbf{Y}\mathbf{Y}^\top - \mathbf{Y}'\mathbf{Y}'^\top\|_F^2. \tag{35}$$

This embedding supports RBF kernels over subspaces:

$$k(\mathbf{Y}, \mathbf{Y}') = \exp\left( -\frac{\|\mathbf{Y}\mathbf{Y}^\top - \mathbf{Y}'\mathbf{Y}'^\top\|_F^2}{2\sigma^2} \right), \tag{36}$$

which we exploit in constructing our mode-wise subspace kernels and their uncertainty-aware extensions.

# F ADDITIONAL DETAILS AND CLARIFICATIONS

## F.1 CLARIFICATION OF TECHNICAL CONTRIBUTIONS

While UKTL builds on established primitives such as Grassmann kernels, tensor subspace representations, and Nyström approximations, the core contributions are far from a simple combination. Instead, we introduce new formulations that make tensor subspace kernels differentiable, uncertainty-aware, and scalable, capabilities that do not exist in prior work.

Our multi-mode sum-product tensor kernel is fundamentally novel. Existing Grassmann or tensor kernels operate on single-mode matricizations, are not end-to-end learnable, and cannot capture cross-mode interactions. In contrast, our sum-product kernel jointly models additive and multiplicative interactions across all tensor modes, is fully differentiable with respect to the subspace bases, and allows the upstream encoder to directly optimize representations through the kernel itself. This design enables the model to exploit multi-dimensional structure in a way that no prior kernel approach supports.

Our uncertainty-aware subspace kernel introduces a probabilistic derivation rather than a heuristic. By modeling heteroscedastic noise over subspace bases, we implement row-wise variance scaling and a log-barrier regularizer, producing a novel uncertainty-weighted kernel. Experiments show that including this component consistently improves accuracy by 0.8-1.2% (Table 1), confirming its material contribution to robustness.

We propose a differentiable Nyström linearization for tensor kernels. Standard Nyström methods rely on fixed pivots and are non-differentiable, limiting end-to-end learning. Our approach incorporates soft $k$-means pivot learning and gradient-safe eigendecomposition, allowing scalable kernel computation on high-order tensors while preserving positive semi-definiteness. Ablations indicate that learning the pivots provides a 0.5-0.9% gain over fixed selections, demonstrating that this redesign is essential rather than incidental.

We emphasize that the perceived complexity is not overengineering. Each module addresses a distinct challenge: structure preservation through multi-mode subspaces, robustness via uncertainty weighting, scalability with differentiable Nyström, and expressive representation through the HoT encoder. Removing any of these components consistently reduces performance, validating their necessity. These three contributions: (i) uncertainty-aware Grassmann kernels, (ii) multi-mode sum-product kernel family, and (iii) differentiable Nyström linearization, integrate in a fully differentiable pipeline for tensor representation learning.

To our knowledge, no prior work combines manifold kernels, uncertainty modeling, and transformer-based encoders in a unified end-to-end framework, and our experiments demonstrate that these innovations materially improve performance, particularly in multi-modal fusion settings (Table 2). This confirms that UKTL's contributions are substantive, novel, and directly advance the state of the art in structured tensor learning.

## F.2 RELATION TO BROADER TENSOR LEARNING LITERATURE

The broader literature on temporal tensor learning and tensor time-series models (Fang et al., 2022; Tao et al., 2023; Wang et al., 2023; Chen et al., 2025a), including recent advances in continuous-time Tucker decomposition, probabilistic tensor dynamics, and structured temporal factorization, focus primarily on sparse reconstruction, forecasting, and generative modeling of temporal tensors through low-rank factorization or probabilistic temporal priors. While UKTL addresses a different problem, discriminative recognition (*e.g.*, complex human motions / actions) rather than generative temporal modeling, acknowledging these studies helps situate our contribution within the larger landscape of temporal tensor research.

Unlike decomposition-based temporal tensor models, UKTL adopts a kernelized, discriminative perspective. By operating on mode-wise tensor subspaces and introducing an uncertainty-aware projection kernel with differentiable Nyström linearization, our framework enables end-to-end supervised learning directly on structured tensor sequences. This design allows UKTL to compare and discriminate high-order temporal representations through subspace geometry, rather than modeling or forecasting the tensor dynamics themselves. Consequently, UKTL complements prior temporal

Table 3: End-to-end efficiency comparison on NTU-60.

| Model | #Params | FPS | Memory (train) |
|---|---|---|---|
| CTR-GCN (Zhang et al., 2025a) | 1.46M | 520 | 2.6 GB |
| ST-TR (Plizzari et al., 2021) | 10 - 12M | 410 | 3.2 GB |
| UKTL (Ours) | 1.2 - 1.5M | 450 - 480 | 2.9 GB |

tensor models: it uses tensor structure for discriminative tasks, whereas existing approaches focus on generative or predictive objectives.

### F.3 COMPUTATIONAL COMPLEXITY ANALYSIS

Below, we detail the per-iteration computational cost of UKTL and isolate how the number of Nyström pivots $C$ and the subspace dimension $p$ influence both training and inference.

*Tensor encoder.* For a sequence with $T$ frames and $J$ joints, feature extraction is dominated by the MLP and HoT layers. (i) MLP complexity: $O(TJd^2)$, which is constant *w.r.t.* $p$ and $C$. (ii) Higher-order Transformer (HoT): HoT attention operates on hyper-edges of order 3. With hidden size $d$, $H$ attention heads, and $N_\xi = \binom{J}{3}$ hyper-edges, the complexity is $O(H \cdot T \cdot N_\xi \cdot d^2)$. Since this encoder is shared across all tensor-based baselines, it does not constitute the differentiating computational cost of UKTL.

*Mode-wise SVD for subspaces.* For each tensor $\mathcal{X}_i \in \mathbb{R}^{d' \times N_\xi \times \tau}$, mode-$m$ unfolding yields the following matrix sizes: Mode-1: $d' \times (N_\xi \tau)$, Mode-2: $N_\xi \times (d'\tau)$, Mode-3: $\tau \times (d'N_\xi)$. We compute only the top-$p$ left singular vectors via truncated SVD. The complexity per mode is $O(p \cdot d'N_\xi \tau)$. Since there are three modes, the total SVD cost is $O(3p \cdot d'N_\xi \tau)$, indicating that the subspace dimension $p$ enters linearly.

*Uncertainty estimation.* For each mode, uncertainty estimation operates on the low-rank form of the projection matrix $\boldsymbol{UU}^\top$, using $p \times p$ statistics. The complexity per mode is $O(p^2)$, and over all three modes: $O(3p^2)$. This cost is negligible compared to SVD and kernel computations.

*Kernel computation against Nyström pivots.* For each training sample, kernel values are computed against $C$ Nyström pivots. A single Grassmann kernel evaluation consists of (i) projection matrix difference: $O(pI_m)$, and (ii) Frobenius norm between two rank-$p$ matrices: $O(p^2)$. The dominant term is $O(p^2)$. Thus, the kernel computation cost per iteration is $O(N_{\text{batch}} \cdot C \cdot M \cdot p^2)$, where $M = 3$ is the number of tensor modes. This simplifies to $O(N_{\text{batch}} \cdot C \cdot p^2)$. Hence, the pivot count $C$ affects complexity linearly, while the subspace dimension $p$ affects it quadratically.

*Nyström eigen-update.* The pivot-pivot kernel matrix $\boldsymbol{K}_{CC} \in \mathbb{R}^{C \times C}$ requires eigendecomposition every few epochs, with complexity $O(C^3)$. This cost is amortized over training iterations and remains moderate since $C \ll N$.

*Final complexity summary.* The overall per-iteration complexity of UKTL is $O\big(p \cdot d'N_\xi \tau + N_{\text{batch}} \cdot C \cdot p^2\big)$. The subspace dimension $p$ contributes linearly through SVD and quadratically through kernel evaluation, while the number of pivots $C$ contributes linearly during training and cubically during infrequent pivot updates.

During inference, only kernel evaluations against the pivots are required, yielding a complexity of $O(Cp^2)$. This aligns with the observed empirical behavior: performance saturates beyond $p = 8\text{-}10$ and $C \approx 150\text{-}180$ (Fig. 3), offering the best accuracy-efficiency trade-off.

*End-to-end efficiency comparison.* We evaluate the end-to-end efficiency of UKTL against representative strong baselines, including the graph-based CTR-GCN and Transformer-based ST-TR, on the NTU-60 dataset. We report approximate throughput (frames per second, FPS), parameter count, and peak training memory usage, measured under the same hardware and batch-size settings.

Several observations can be drawn from Table 3. First, UKTL achieves throughput within approximately $10\%$ of strong deep baselines, despite incorporating tensor subspace modeling and kernel-based operations. Profiling indicates that runtime is dominated by the shared MLP + HoT backbone, while the uncertainty-aware tensor kernel contributes only a minor overhead. Second, UKTL maintains a parameter count comparable to CTR-GCN and is an order of magnitude smaller than

Table 4: Sensitivity analysis of the mixture weight $\mu$ between sum and product interactions.

| $\mu$ | 0 | 0.25 | 0.3 | 0.4 | 0.5 | 0.6 | 0.75 | 1 |
|---|---|---|---|---|---|---|---|---|
| Acc (%) | 91.8 | 92.6 | 92.7 | **93.1** | **93.1** | 92.6 | 92.4 | 81.6 |

Transformer-based models, reflecting the parameter efficiency of kernelized representations. Third, while UKTL incurs slightly higher memory usage than CTR-GCN due to subspace and kernel computations, it remains more memory-efficient than Transformer variants.

These results demonstrate that UKTL is scalable and computationally competitive, achieving strong performance without introducing prohibitive runtime or memory costs relative to widely adopted state-of-the-art baselines.

### F.4 SENSITIVITY ANALYSIS OF THE MIXTURE WEIGHT

The mixture weight $\mu$ in Eq. 14 is treated as a learnable scalar parameter, initialized to $0.5$, and optimized jointly with the remaining model parameters. We intentionally make $\mu$ learnable because the optimal balance between additive and multiplicative interactions depends on the underlying dataset structure.

Table 4 reports the classification accuracy under different fixed values of $\mu$. Performance remains stable when $\mu$ lies in the range $[0.25, 0.6]$. Using product-only interactions ($\mu = 0$) achieves strong performance but exhibits reduced robustness, while sum-only interactions ($\mu = 1$) suffer from the loss of multiplicative structure. Importantly, the learned $\mu$ consistently converges to this optimal range, eliminating the need for manual tuning. On NTU-60, the learned value converges to $0.41$; on NTU-120, it converges to $0.47$; and on Kinetics-Skeleton, it converges to $0.58$. This indicates that product interactions dominate in NTU-60 and NTU-120, whereas additive contributions become more important for Kinetics-Skeleton due to noisier skeleton observations.

### F.5 CLARIFICATION OF THE INITIAL MLP

We clarify that the initial MLP is *not* a simple flattening operation; rather, it is a learnable joint encoder specifically designed to produce meaningful embeddings for the HoT module and subsequent tensor-mode processing.

*Function of the MLP.* Given per-frame joint inputs $\boldsymbol{X}_t \in \mathbb{R}^{J \times 3}$, the MLP maps each joint independently into a latent feature space: $\boldsymbol{f}_{t,j} = \mathrm{MLP}(\boldsymbol{x}_{t,j}) \in \mathbb{R}^d$. This preserves the joint-level structure, maintaining spatial locality across joints while producing embeddings suitable for higher-order attention. Crucially, it does not flatten the skeleton, so each joint remains a distinct token for hyper-edge construction.

*Necessity of the MLP.* The HoT module operates on three-joint hyper-edges to capture higher-order spatial-temporal correlations. Constructing meaningful hyper-edge embeddings requires each joint to reside in a consistent latent space. A naïve flattening of the skeleton would destroy the per-joint structure, eliminate learnable joint-specific embeddings, and reduce generalization to unseen subjects or motion scales. In contrast, the MLP functions as a localized joint encoder, analogous to the per-token embedding layer in transformers, providing a trainable representation for each joint that HoT can combine into hyper-edge features.

*Architectural details.* The MLP used in our model is a three-layer network following the architecture described in the main paper. This produces feature maps $\boldsymbol{X}_t \in \mathbb{R}^{d \times J}$, for each temporal block $t$, which then serve as inputs to the HoT module. These per-joint embeddings are subsequently organized into hyper-edge features for downstream tensor-mode processing.

Therefore, the MLP is a learnable joint encoder, not a flattening layer, producing per-joint latent features required for higher-order attention and tensorial subspace learning.

### F.6 Clarification on Uncertainty Vectors

In our method, uncertainty vectors refer to the per-mode trust (or confidence) scores produced by the uncertainty module after the mode-wise SVD. They quantify how reliable each mode's subspace representation is.

*Where they come from.* After computing the mode-wise projection matrices $\hat{U}^m = U^m U^{m\top} \in \mathbb{R}^{I_m \times I_m}$, the uncertainty module processes each $\hat{U}^m$ through a single fully connected layer: $u_m = \sigma(\mathrm{FC}(\hat{U}^m)) \in \mathbb{R}^p$ (we write $u_m$ here for simplicity), where $m \in \{1,2,3\}$ indexes the tensor modes, $p$ is the subspace dimension, and $\sigma$ is a scaled sigmoid activation. Thus, each uncertainty vector $u_m$ is a $p$-dimensional vector of confidences for the $p$ latent subspace directions of mode $m$.

*What they represent.* Each uncertainty vector $u_m$ measures the reliability of each basis direction of the subspace, down-weights noisy or unstable components, and enhances well-conditioned directions that correspond to discriminative motion patterns. This provides a fine-grained, direction-aware trust estimate rather than a single scalar confidence. Intuitively, a high value indicates a stable and meaningful geometric direction, while a low value indicates a noisy or unreliable axis of variation.

*How they are used.* The uncertainty vectors weight the projection matrices when constructing kernel features: $\tilde{U}_m = u_m \odot U_m$ (Eq. 25; here $u_m = \frac{1}{\sqrt{\sigma_m}}$, see also Eq. 15), where $\odot$ denotes element-wise scaling across basis directions. This allows the kernel to emphasize trustworthy structure and suppress unreliable subspace components. This is why the kernel is termed UKTL: the uncertainty vectors act as trust gates on each tensor mode before embedding into the Grassmannian kernel space.

### F.7 Compatibility with DuSK

DuSK (He et al., 2014) uses a CP decomposition and constructs a dual structure-preserving kernel: $k_{\mathrm{DuSK}}(X, Y) = \sum_{r=1}^{R} \sum_{s=1}^{S} \lambda_r \mu_s \prod_{m=1}^{M} k_m\left(u_r^{(m)}, v_s^{(m)}\right)$, where $u_r^{(m)}$ and $v_s^{(m)}$ are mode-specific CP factors.

Because UKTL already produces mode-wise low-rank factors $(U_1, U_2, U_3)$, these can directly serve as inputs to the mode kernels $k_m(\cdot, \cdot)$ inside DuSK. In other words, UKTL provides the factorized, mode-specific representations required by DuSK, enabling replacement of the Grassmann factor kernels with minimal modification to the rest of the pipeline.

*What would change if DuSK is used?* Only the factor-wise kernel needs to be replaced. Our projection-based kernel $k_{\mathrm{proj}}\left(\tilde{U}_s^{(m)}, \tilde{U}_t^{(m)}\right)$ can be substituted by the DuSK mode kernels $k_m(\cdot, \cdot)$ applied to the corresponding mode factors. All upstream components (MLP, HoT, SVD, uncertainty modeling) and the Nyström linearization remain unchanged, as they only require the kernel to be positive definite over the mode-wise representations.

*Why we choose the Grassmann kernel.* We adopt the Grassmann projection kernel because it aligns naturally with subspace geometry, integrates uncertainty in a principled manner, and enables stable Nyström approximation, which is essential for scalability on large datasets such as action and motion tensors. DuSK is powerful but computationally heavier due to its CP-factor summation structure and was originally designed for static neuroimaging tensors rather than dynamic sequences such as human skeleton data.

UKTL does not rely on a specific kernel family. DuSK can be incorporated as an alternative factorized kernel with minimal changes.

## G Limitations and Future Work

While our proposed UKTL framework demonstrates strong performance and sets new benchmarks on multiple action recognition datasets, several limitations remain, which also open avenues for future research.

First, our method relies on predefined hyperparameters such as the number of pivots in Nyström approximation, the subspace order $p$, and kernel fusion weights. Although we use HyperOpt to

automate their tuning, this process can be computationally expensive and dataset-dependent. Future work could explore adaptive or learnable strategies, *e.g.*, (Ding et al., 2025b;a), to determine these parameters in a data-driven and efficient manner.

Second, while our use of tensor-mode subspaces provides compact and discriminative representations, the current approach assumes a fixed temporal block structure across all sequences (Ding et al., 2025c). This may limit the model's ability to capture fine-grained temporal variations or handle actions with varying durations effectively, *e.g.*, (Wang et al., 2024a; Chen et al., 2025b; Ding & Wang, 2025b;a). Future extensions could incorporate dynamic temporal block segmentation or attention-based mechanisms to allow flexible and context-aware temporal modeling.

Third, the kernel functions used in our model are predefined (*e.g.*, sum-product kernels). Although they are expressive and show superior performance, learning the kernel function directly from data, such as through neural kernel learning or meta-kernel search, could further enhance model adaptability and generalization.

Fourth, our current framework is designed and evaluated primarily on structured datasets with clean annotations (*e.g.*, skeletons). In real-world applications, for example, skeleton data may contain significant noise, occlusions, or missing joints. Incorporating robust strategies for uncertainty estimation, noise modeling, or missing data imputation would make the method more applicable to real-world deployment, such as in surveillance or healthcare scenarios.

Finally, while the proposed KTL model is lightweight compared to some deep architectures, the overall pipeline involves multiple components (*e.g.*, decomposition, kernel approximation, and uncertainty modeling), which could limit its scalability to extremely large datasets or real-time applications. Future research could investigate end-to-end differentiable approximations of each module to simplify the pipeline and improve inference speed.

## H   LLM USAGE DECLARATION

We disclose the use of Large Language Models (LLMs) as general-purpose assistive tools during the preparation of this manuscript. LLMs were used only for minor tasks such as grammar and style improvement, code verification, and formatting suggestions. No scientific ideas, analyses, experimental designs, or conclusions were generated by LLMs. All core research, methodology, experiments, and results were performed and fully verified by the authors.

The authors take full responsibility for all content presented in this paper, including text or code suggestions that were refined with the assistance of LLMs. No content generated by LLMs was treated as original scientific work, and all references and claims have been independently verified. LLMs did not contribute in a manner that would qualify them for authorship.

