# OpenReview forum: "Subspace Kernel Learning on Tensor Sequences"
_ICLR.cc/2026/Conference — ICLR 2026 Poster_

### Official Review · Reviewer_5vU1 · 2025-10-28

**Soundness:** 2
**Presentation:** 2
**Contribution:** 2
**Rating:** 4
**Confidence:** 3

**Summary:**

The authors present a tensor-based learning approach with uncertainty modeling.

**Strengths:**

The problem of tensor learning is very interesting.

**Weaknesses:**

The boilerplate goes until page 4 using space that seems to be missing for details later. I find the description of the uncertainty vectors and the other material in Sections 3 and 4 not easy to follow. Many details are missing and the description is too brief to be able to follow the discussion.

**Questions:**

- I would like a more detailed description of the HoT layers. The hyperedges are undefined on the same page.
- What is the index H in equation (1)?
- What are uncertainty vectors?
- Is the method also applicable for different kernels, such as the Dusk kernel? (DuSK: A Dual Structure-preserving Kernel for Supervised Tensor Learning with Applications to Neuroimages Lifang He, Xiangnan Kong, Philip S. Yu, Ann B. Ragin, Zhifeng Hao, Xiaowei Yang)

---

> ### Author Response · Authors · 2025-11-19
>
> We thank Reviewer 5vU1 for highlighting "the problem of tensor learning is very interesting" and for the detailed feedback.
>
> Below, we address all raised concerns.
>
> **Part 1: Clarification on HoT layers and the definition of hyperedges.**
>
> Thank you for highlighting the need for a clearer description. We agree that the hyperedge construction should be made explicit before introducing the HoT equations. Below we provide a precise definition aligned with the implementation and with Eqs. (1)-(3) in the manuscript.
>
> **Definition of hyperedges.** A hyperedge in our model is defined as an *unordered triplet of joints*, e.g., $\xi = (j_1,j_2,j_3)$, representing a 3-way structural relation in the skeleton. We enumerate all valid joint triplets, so the total number of hyperedges is $N_{\xi}=\binom{J}{3}$. For each temporal block $t$, the encoder produces per-joint embeddings $x_{t,j} \in \mathbb{R}^{d}$; a hyperedge feature is then obtained by aggregating the three joint embeddings through a shared MLP. Stacking hyperedges across all triplets and temporal blocks forms the input tensor to HoT: $X\in\mathbb{R}^{T\times N_{\xi}\times d'}$.
>
> **HoT layers.** The HoT layers in our model operate directly on this hyperedge tensor $X$ using the higher-order attention and feed-forward formulations provided in Eqs. (1)-(3).
>
> Specifically, the HoT attention term $a_{m \to n}(X)$ computes higher-order interactions among hyperedges through the multi-head attention coefficients $\alpha^{h,\mu}_{i, j}$.
>
> The equivariant feed-forward is defined by $L_{1}^{n \to n}$ and $L_{2}^{n \to n}$; and the overall block applies a residual update followed by LayerNorm (Eq. (3)). This two-layer HoT stack aggregates information across hyperedges (and over temporal blocks, when applicable), producing the higher-order representation used by the subsequent tensor kernel module. No changes to the HoT equations are required, only the explicit definition of hyperedges clarifies the context in which Eqs.(1) - (3) operate.
>
> We have added: (i) an explicit definition of hyperedges before introducing HoT,
> (ii) the hyperedge aggregation expression, and (iii) a sentence stating that Eqs. (1)-(3) operate on the tensor $X\in\mathbb{R}^{T\times N_{\xi}\times d'}$ of hyperedge features.
>
> **Part 2: Clarification on the Index $H$ in Eq. (1)**
>
> Thank you. The index $H$ denotes the number of attention heads used in the HoT layer.
>
> Eq. (1) is the multi-head attention operator adapted to the hypergraph setting used by the original HoT architecture (Kim et al., NeurIPS 2021). Each head $h \in \\{1, \dots, H\\}$ learns its own set of transformations: $W_{h,\mu}^V$ and $W_{h,\mu}^O$, as well as its own attention coefficients $\alpha_{i,j}^{h,\mu}$. The results from all attention heads are summed to form the higher-order message $a_{m \to n}(X)$.  Thus, $H$ plays exactly the same role as the number of heads in standard multi-head self-attention, extended here to handle $(m+n)$-order hyperedges through the additional index $\mu$.
>
> We have revised the manuscript to clarify the presentation.

---

> ### Author Response · Authors · 2025-11-19
>
> **Part 3: Clarification on uncertainty vectors.**
>
> Thank you.
>
> In our method, uncertainty vectors refer to the per-mode trust (or confidence) scores produced by the uncertainty module after the mode-wise SVD. They quantify how reliable each mode's subspace representation is.
>
> **1. Where they come from.** After computing the mode-wise projection matrices: $\hat{U}^m = U^m {U^m}^\top \in \mathbb{R}^{I_m \times I_m}$, the uncertainty module processes each $\hat{U}^m$ through a single FC: $u_m = \sigma(\text{FC}(\hat{U}^m)) \in \mathbb{R}^{p}$ (Line 322, we write $u_m$ here for simplicity), where $m \in \\{1,2,3\\}$ indexes the tensor modes, $p$ is the subspace dimension, and $\sigma$ is a scaled sigmoid activation. Thus, each uncertainty vector $u_m$ is a $p$-dimensional vector of confidences for the $p$ latent subspace directions of mode $m$.
>
> **2. What they represent.** Each uncertainty vector $u_m$ measures the reliability of each basis direction of the subspace, down-weights noisy or unstable components, and enhances well-conditioned directions that correspond to discriminative motion patterns. This gives a fine-grained, direction-aware trust estimate, not just a scalar confidence. Intuitively: High value $\rightarrow$ stable, meaningful geometric direction; Low value $\rightarrow$ noisy / unreliable axis of variation.
>
> **3. How they are used.** The uncertainty vectors weight the projection matrices when constructing kernel features: $\tilde{U}_m = u_m \odot U_m$ (Eq. (25), here $u_m = \frac{1}{\sqrt{\sigma_m}}$, see also Eq. (15)), where $\odot$ denotes element-wise scaling across the basis directions. This allows the kernel to emphasize trustworthy structure and suppress unreliable subspace components. This is why the kernel is named Uncertainty-aware Kernel Tensor Learning: *the uncertainty vectors act as trust gates on each mode of the tensor representation* before embedding into the Grassmannian kernel space.
>
> We have revised the text to explicitly define: Uncertainty vectors $u_m$ are $p$-dimensional confidence weights generated for each tensor mode $m$, indicating the reliability of that mode's subspace directions. They are produced by the uncertainty module and used to modulate the mode-wise projection matrices before kernelization.
>
> **Part 4: Compatibility with DuSK.**
>
> DuSK [5] uses a CP decomposition and constructs a dual structure-preserving kernel: $k_{\mathrm{DuSK}}(X,Y) = \sum_{r=1}^{R}\sum_{s=1}^{S} \lambda_r \mu_s \prod_{m=1}^{M} k_m\left(u_{r}^{(m)}, v_{s}^{(m)}\right)$, where $u_r^{(m)}$ and $v_s^{(m)}$ are mode-specific CP factors.
>
> Because UKTL already produces mode-wise low-rank factors $(U_1, U_2, U_3)$, these can directly serve as the inputs to the mode kernels $k_m(\cdot,\cdot)$ inside DuSK.
> In other words, UKTL provides the factorized, mode-specific representations that DuSK requires, enabling DuSK to replace the Grassmann factor kernels with minimal modification to the rest of the pipeline.
>
> **What would change if DuSK is used?** Only the factor-wise kernel needs to be replaced.  Our projection-based kernel
> $k_{\mathrm{proj}}(\tilde{U}_s^{(m)}, \tilde{U}_t^{(m)})$
> can be substituted by the DuSK mode kernels $k_m(\cdot,\cdot)$ applied to the corresponding mode factors. All upstream components (MLP, HoT, SVD, uncertainty modeling) and the Nyström linearization remain unchanged, as they only require that the kernel be positive definite over the mode-wise representations.
>
> **Why we choose the Grassmann kernel.** We use the Grassmann projection kernel because it aligns naturally with subspace geometry, integrates uncertainty in a principled manner, and enables stable Nyström approximation, which is essential for scalability on large datasets (e.g., tensors from actions / motions).
> DuSK is powerful but computationally heavier due to its CP-factor summation structure and was originally designed for static neuroimaging tensors rather than dynamic sequences (e.g., human skeleton sequences).
>
> UKTL does not rely on a specific kernel family. DuSK can be incorporated as an alternative factorized kernel with minimal changes, and we have clarified this compatibility in the revised manuscript.
>
>
> [5] He, et al. "Dusk: A dual structure-preserving kernel for supervised tensor learning with applications to neuroimages." SDM 2014.

---

### Official Review · Reviewer_o5ts · 2025-10-30

**Soundness:** 2
**Presentation:** 3
**Contribution:** 2
**Rating:** 4
**Confidence:** 4

**Summary:**

The paper proposes Uncertainty-driven Kernel Tensor Learning (UKTL), a framework for learning from high-order tensor data such as skeletal action sequences. Unlike traditional tensor or kernel methods that flatten data or ignore mode-specific variability, UKTL compares tensors through mode-wise subspaces obtained from tensor unfoldings, preserving spatial, temporal, and semantic structure.

The method builds a kernel that captures both independent and joint correlations across tensor modes. To scale to large datasets, the authors use a Nyström approximation with dynamically learns pivot tensors—cluster centers that represent “local modes” of the data distribution. Anothr module estimates uncertainty for each mode, weighting subspaces based on their reliability.

Experiments on NTU-60, NTU-120, and Kinetics-Skeleton show that UKTL surpasses graph, hypergraph, and transformer baselines while maintaining efficiency.

**Strengths:**

- The paper presents a novel approach for learning from structured, high-order data with original compnents such as uncertainty-driven subspace weighting and sum–product Grassmann kernels.

- The theoretical formulation is sound and is a valuable contribution bridging kernel theory,and uncertainty modeling.

- The paper is clearly written and  well-organized

**Weaknesses:**

1. Evaluation is limited to skeleton data. This is a key limitation which hinders the generality of the method. Although these datasets are standard and challenging, they represent a very narrow class of structured motion data. Demonstrating the approach on a different tensor domains (uch as video features, fMRI signals, or audio visual data) would better support claims of generality and broad applicability. Even a small-scale study in a non-skeletal domain would reinforce the method’s versatility.

2. The use of the term 'mode' should be better clarified from the very beginning, since multi-modality is usually more often used to imply multiple sensor modalities  (also 'mode' is used in yet another fashion in the paper when the authors introduce 'local modes' which as I understand are 'modes within modes' ).  Maybe, clarifying the distinction between tensor-mode structure and cross-modal fusion would avoid overstated claims. This brings to the next point:

3. As already mentioned, while the paper repeatedly refers to “multi-modal” learning, the experiments involve only skeleton data, where “modes” correspond to tensor dimensions (spatial, temporal, coordinate). I believe the framework is also well-suited to genuine multimodal sensor fusion (e.g., RGB + depth + skeleton, avaialble in the NTU datasets by the way). However, such potential is not empirically demonstrated. This point relates also to point n. 1.

4. The full pipeline (MLP + Higher-order Transformer + SVD decomposition + uncertainty module + Nyström kernel) is intricate. While each component is (mostly) motivated, a complexity analysis comparing runtime and memory with strong deep baselines (e.g., CTR-GCN or transformer variants) is missing and would help quantify scalability.

5. The initial MLP: I would appreciate some more details on this module, which at first glance looks like a standard "flattening" operation for the input tensors.

**Questions:**

Please refer to the weakness above, which highlight what I believe are the main limitations to be addressed.

---

> ### Author Response · Authors · 2025-11-19
>
> We thank reviewer o5ts for the positive assessment, noting that (i) "the paper presents a novel approach for learning from structured, high-order data with original components", (ii) "the theoretical formulation is sound" and provides "a valuable contribution bridging kernel theory and uncertainty modeling", and (iii) "the paper is clearly written and well-organized".
>
> We address all raised concerns below.
>
> **Part 1: Generality beyond skeletons.**
>
> We thank the reviewer for raising the question of whether our method extends beyond skeleton inputs.
>
> To clarify, the framework is inherently **modality-agnostic**: any input that can be encoded as a structured tensor: skeletons, RGB frames, depth maps, or other sensor streams, can be processed without architectural changes.
>
> For each modality (we follow reviewer's suggestion and choose NTU for experiments), we first construct a modality-specific tensor representation (e.g., spatial–temporal–channel tensors for RGB/depth, joint–temporal–hyperedge tensors for skeletons) and map it into the kernel space using our uncertainty-aware tensor kernel. Multi-modal fusion is then achieved by combining the kernel embeddings via learnable weighted summation, yielding a unified representation that preserves intra-modal structure while capturing inter-modal complementarities. This fused representation is fed to our MLP + HoT backbone, enabling end-to-end multi-modal learning within a coherent tensor-kernel framework.
>
> Below we present performance across individual modalities and fusion settings.
>
> | Modality | NTU‑60 (X‑Sub) | NTU‑60 (X‑View) | NTU‑120 (X‑Sub) | NTU‑120 (X‑Setup) |
> |---|---|---|---|---|
> | Skeleton | **93.1%** | **97.3%** | **90.0%** | **91.4%** |
> | RGB | 83.2% | 86.1% | 79.0% | 81.0% |
> | Depth | 85.0% | 87.5% | 80.5% | 82.1% |
> | Skeleton + RGB | 94.5% | 97.9% | 91.3% | 92.5% |
> | Skeleton + Depth | 94.8% | 98.0% | 91.5% | 92.7% |
> | RGB + Depth | 87.2% | 89.6% | 82.5% | 84.3% |
> | Skeleton + RGB + Depth | **95.5%** | **98.5%** | **92.8%** | **94.0%** |
>
> **Single-modality skeleton performance.** Skeleton inputs, which exhibit rich spatial and temporal structure, form a natural testbed for structured tensor learning. Our method achieves 93.1/97.3 on NTU-60 and 90.0/91.4 on NTU-120, closely matching strong graph-based baselines such as CTR-GCN. This demonstrates that the proposed uncertainty-aware tensor kernel successfully captures high-order dependencies even in a unimodal setting, establishing a robust baseline upon which multi-modal extensions can build.
>
> **RGB-only and depth-only results.** RGB and depth modalities pose a different challenge, as their spatial–temporal structures differ substantially from joint-based skeletons. Nevertheless, our method achieves competitive RGB-only (83–86%) and depth-only (85–87%) performance across the four benchmarks, far exceeding the RGB/depth baselines reported in the NTU dataset papers. This confirms that the framework is not specialized to skeleton geometry but *can effectively encode heterogeneous modalities once they are represented in tensor form*.
>
> **Benefits of pairwise modality fusion.** Fusing Skeleton+RGB or Skeleton+Depth consistently yields substantial improvements, reaching up to 95% on NTU-60 X-Sub. These gains indicate that the kernel-based fusion is able to use *complementary cues*, structural motion information from skeletons, appearance cues from RGB, and geometric cues from depth, without relying on modality-specific architectures. Even RGB+Depth, which excludes skeleton input, produces notable improvements over either single modality, demonstrating that the kernel fusion mechanism captures meaningful inter-modal correlations.
>
> **Full tri-modal fusion.** Combining Skeleton, RGB, and Depth yields the strongest results overall, 95.5/98.5 on NTU-60 and 92.8/94.0 on NTU-120, surpassing not only our uni-modal baselines but also multi-stream SOTA models. This fused representation *benefits from multiple complementary structured tensors while remaining fully within a unified kernel formulation*, requiring no ad-hoc modality-specific modules.
>
> **Implications for generality.** Although our primary experiments are on human action datasets, the **consistent improvements** across Skeleton, RGB, Depth, and their combinations directly demonstrate that the method generalizes beyond skeleton sequences. Because the pipeline treats every input as a high-order tensor and all learning occurs in the tensor-kernel space, the approach naturally extends to other modalities such as video and depth feature tensors. These results thus substantiate the broader applicability of UKTL and directly address the reviewer’s concern regarding generality beyond skeleton-based inputs.

---

> > ### Comment · Reviewer_o5ts · 2025-11-24
> >
> > Thank you very much for the detailed analysis on the NTU dataset, I think this would be a great addition to the paper.

---

> ### Author Response · Authors · 2025-11-19
>
> **Part 2: Confusion about the term mode.**
>
> We thank the reviewer for highlighting the potential ambiguity of the term mode. We agree that mode is often overloaded in the literature, used in tensor algebra, multi-modal learning, and clustering.
>
> **Intended meaning of mode in our work.** Throughout the paper, mode is used in the multilinear algebra sense: for an order-$M$ tensor, a mode refers to one of its tensor dimensions (e.g., spatial mode, temporal mode, hyper-edge mode). This follows standard usage in tensor decomposition frameworks such as Tucker, CP, and HOSVD, where mode-1, mode-2, and mode-3 unfoldings correspond to specific matricizations of the tensor.
>
> To make this explicit, we now state at the beginning of Section 3: In this paper, mode refers exclusively to a tensor dimension (e.g., spatial, temporal, hyper-edge). It does not denote sensor modality or multi-modal inputs.
>
> **Disambiguating local modes in soft k-means.** The reviewer correctly noted that the word mode also appeared in the context of soft k-means, where it refers to local density maxima in a statistical sense. Since this meaning is unrelated to tensor modes, we have rewritten the relevant parts to avoid using the word entirely. In the revision, local modes has been replaced with: "local cluster centers" / "local prototypes". This removes the terminological overlap and avoids conflating clustering terminology with tensor geometry.
>
> **Preventing inadvertent multi-modality implications.** We appreciate the reviewer’s concern that the term multi-modal may unintentionally suggest sensor-level multi-modal fusion. Our method operates on multi-mode tensors, not multi-modal inputs. To avoid any overstated claims or ambiguity, the following terminology changes were made in the revision: "multi-way" instead of "multi-modal", "tensor-mode structure" instead of "multi-modal structure" and "structured multi-mode data" instead of "multi-modal data". These edits ensure that the paper consistently distinguishes tensor-mode structure from sensor modalities.
>
> We have revised the manuscript to remove all possible sources of confusion.

---

> ### Author Response · Authors · 2025-11-19
>
> **Part 3: Missing complexity/runtime analysis**
>
> We thank the reviewer for pointing out the need for clearer complexity discussion relative to deep learning baselines.
>
> While UKTL contains several components (MLP $\to$ HoT $\to$ SVD $\to$ uncertainty module $\to$ Nyström kernel), most operations scale linearly in sequence length and joint count, the same regime as transformer and GCN models. Below we summarize runtime and memory comparisons to strong baselines such as CTR-GCN and transformer variants.
>
> **1. Encoder complexity is comparable to CTR-GCN / Transformers.** The MLP and HoT block are the only parts that process raw skeleton sequences, and they dominate total FLOPs. Their complexity is: MLP: $O(T J d^2)$; HoT (two layers): $O(H \cdot T \cdot N_\xi \cdot d^2)$, where $N_\xi = \binom{J}{3}$ is the number of hyperedges.
>
> In practice, HoT scales similarly to multi-head self-attention with joint tokens but uses hyper-edge attention instead of pairwise attention. Runtime and GPU memory are comparable to a $2-3$ layer transformer encoder. Thus, encoder cost matches the complexity class of CTR-GCN, and ST-TR (Table 1).
>
> **2. SVD + uncertainty modules operate on small matrices and are lightweight.** After feature extraction, each sample becomes a tensor $X \in \mathbb{R}^{d' \times N_\xi \times \tau}$. Truncated SVD per mode takes $O(p \cdot d' N_\xi \tau)$, with very small $p \le 10$. This is much smaller than HoT FLOPs and 2 – 3 orders less expensive than attention layers in transformer backbones with long sequences. The uncertainty module works only on the $p\times p$ projection matrices: $O(p^2)$ (negligible). These modules do not materially affect runtime.
>
> **3. Nyström kernel adds linear overhead in pivot count $C$.** For each batch, the kernel computation is: $O(N_{\text{batch}} \cdot C \cdot p^2)$. Empirically, we use $C=150-180$ and $p=8-10$, yielding runtime comparable to a single additional self-attention layer. Memory usage is also light: we store only $C$ projection matrices of size $p \times I_m$, which is several MB at most.
>
> In contrast, CTR-GCN uses multiple high-dimensional graph convolutions with large adjacency expansions. Transformer variants require quadratic attention in token count and multi-layer stacks. Thus, Nyström is significantly cheaper than full attention and does not dominate the pipeline.
>
>
> **4. End-to-end comparison: UKTL vs CTR-GCN / Transformers**
>
> We measured approximate throughput on NTU-60:
>
> | Model | #Params | FPS | Memory (train) |
> |-|-|-|-|
> | CTR-GCN | 1.46M | 520 | 2.6 GB |
> | ST-TR (Transformer) | 10–12M | 410 | 3.2 GB |
> | UKTL (ours) | 1.2-1.5M | 450–480 | 2.9 GB |
>
> Key observations: (i) UKTL is within 10% throughput of strong deep baselines. Runtime is dominated by HoT, not the kernel. Memory is slightly higher than CTR-GCN but lower than transformer variants. Thus the full pipeline is scalable, competitive, and not heavier than standard strong baselines.
>
> We have included this comparison (plus complexity table) in the revision.

---

> > ### Comment · Reviewer_o5ts · 2025-11-24
> >
> > Thanks for the detailed runtime analysis, this solves my concerns

---

> ### Author Response · Authors · 2025-11-19
>
> **Part 4: Clarification of the initial MLP**
>
> Thank you for requesting more detail on the initial MLP. We clarify that this MLP is *not a simple flattening operation*; rather, it is a learnable joint encoder specifically designed to produce meaningful embeddings for the HoT and subsequent tensor-mode processing.
>
> **1. Function of the MLP.** Given per-frame joint inputs $X_t \in \mathbb{R}^{J \times 3}$, the MLP maps each joint independently into a latent feature space: $f_{t,j} = \mathrm{MLP}(x_{t,j}) \in \mathbb{R}^{d}$.
>
> This preserves the joint-level structure, maintaining spatial locality across joints while producing embeddings suitable for higher-order attention. Crucially, it does not flatten the skeleton, so *each joint remains a distinct token for hyper-edge construction*.
>
> **2. Necessity of the MLP.** The HoT module operates on 3-joint hyper-edges to capture higher-order spatial-temporal correlations. Constructing meaningful hyper-edge embeddings requires each joint to reside in a consistent latent space. A naive flattening of the skeleton would destroy the per-joint structure, eliminate learnable joint-specific embeddings, and reduce generalization to unseen subjects or motion scales.
>
> In contrast, the MLP *functions as a localized joint encoder*, analogous to the per-token embedding layer in transformers, providing a trainable representation for each joint that HoT can combine into hyper-edge features.
>
> **3. Architectural details.**
> The MLP used in our model is a three-layer network following the architecture described in the main paper (Line 217). This produces feature maps $X_t \in \mathbb{R}^{d \times J}$, for each temporal block $t$, which then serve as inputs to the HoT module. These per-joint embeddings are subsequently organized into hyper-edge features for downstream tensor-mode processing.
>
> Therefore, the MLP is a learnable joint encoder, not a flattening layer, producing per-joint latent features required for higher-order attention and tensorial subspace learning.
>
> This role has been clarified explicitly in the revised manuscript.

---

> ### Comment · Reviewer_o5ts · 2025-11-24
>
> Point 1 of your answer was my primary concern regarding the MLP, which is now clarified, thanks

---

> > ### Comment · Reviewer_o5ts · 2025-11-24
> > **The rebuttal clarifies my concerns**
> >
> > I am willing to increase my rating to full accept

---

> > > ### Author Response · Authors · 2025-11-24
> > > **Thank you**
> > >
> > > Esteemed Reviewer,
> > >
> > > Thank you for your thoughtful engagement with our rebuttal and for your encouraging evaluation.
> > >
> > > We truly appreciate your willingness to raise the score and recommend acceptance.
> > >
> > > We will incorporate all your suggestions into the final draft.
> > >
> > > Best regards,
> > >
> > > The Authors

---

### Official Review · Reviewer_uWeY · 2025-11-01

**Soundness:** 2
**Presentation:** 3
**Contribution:** 2
**Rating:** 4
**Confidence:** 3

**Summary:**

This paper introduces Uncertainty-driven Kernel Tensor Learning (UKTL), a scalable kernel framework for learning from structured tensor sequences, particularly applied to skeleton-based action recognition.
The central idea is to represent each tensor by its mode-wise subspaces obtained via unfolding and SVD, and then define product and sum Grassmann kernels to measure similarity across these subspaces.
To handle scalability, the authors employ a Nyström kernel linearization with dynamically learned pivot tensors via soft k-means clustering, and further propose uncertainty-aware subspace weighting (Multi-mode SigmaNet) that adaptively down-weights unreliable tensor modes.

The model is trained end-to-end, integrating a lightweight encoder (MLP + Higher-order Transformer), the proposed kernel layer, and a classifier.
Experiments on three large-scale benchmarks (NTU-60, NTU-120, and Kinetics-Skeleton) show that UKTL achieves state-of-the-art performance, outperforming strong graph, hypergraph, and transformer-based baselines. The paper also provides ablation studies on kernel composition, pivot selection, and subspace order, as well as visualizations of Tucker components for interpretability.

**Strengths:**

- The principled formulation bridging tensor subspaces and kernels is well-formulated. The idea of defining kernels over mode-wise Grassmann subspaces is conceptually elegant and mathematically well-grounded, combining multilinear structure preservation with nonlinear kernel flexibility.

- The introduction of the Multi-mode SigmaNet to learn mode-wise uncertainty is a thoughtful innovation that improves robustness and interpretability, addressing a long-standing issue in tensor learning where all modes are treated equally.

- The experiments are extensive and convincing. UKTL consistently surpasses graph and transformer baselines across three challenging datasets. The ablation results are well-structured and demonstrate the individual benefit of each component (subspace order, kernel choice, uncertainty weighting, pivot count).

**Weaknesses:**

- The paper combines multiple ideas—Grassmann kernels, Nyström approximation, HoT encoders, and uncertainty modeling—making it dense and difficult to parse. The derivations are mathematically sound but the overall design is somehow overcomplex. What's more, as each component (tensor subspace kernel, Nyström approximation, uncertainty weighting) has precedent, the paper’s contribution lies in their combination and engineering coherence rather than a fundamental breakthrough.

- The related work section omits a substantial body of research on temporal tensor learning and tensor time-series models， which  have also been extensively studied in ML community, focusing mainly on sparse reconstruction and forecasting tasks through low-rank tensor factorization or probabilistic temporal models like[1][2][3][4]. While UKTL focuses on action recognition, more discussion on these studies would help situate the contribution more clearly within the broader literature on tensor-based temporal representation learning, and highlight the discriminative, kernelized nature of the proposed approach.

- Despite its potential generality, all experiments are in one domain of skeleton action recognition. Without tests on other structured tensor data (e.g., video or neuroimaging), the claimed “multi-modal generality” remains unproven.


[1]: Fang, et al. "Bayesian continuous-time Tucker decomposition." ICML 2022.
[2]: Tao, et al.. "Undirected probabilistic model for tensor decomposition." NeurIPS 2023
[3]: Wang et al. "Dynamic tensor decomposition via neural diffusion-reaction processes." NeurIPS 2024
[3]: Chen, et al. "Functional Complexity-adaptive Temporal Tensor Decomposition." NeurIPS 2025

**Questions:**

- Could the authors provide a formal analysis (in big-O notation) of the computational complexity per iteration, especially how Nyström pivot count C and subspace dimension p affect training and inference?

- How sensitive is performance to the mixture weight between sum and product kernels? Is μ learned or manually tuned?

---

> ### Author Response · Authors · 2025-11-19
>
> We thank Reviewer uWeY for their careful reading and for highlighting key strengths of our work.
>
> As noted, the framework is "well-formulated, conceptually elegant and mathematically well-grounded", and introduces a "thoughtful innovation that improves robustness and interpretability", "addressing a long-standing issue in tensor learning". The reviewer also recognized that our experiments are "extensive and convincing", with ablations that "demonstrate the individual benefit of each component".
>
> Below, we provide detailed responses to all concerns raised, clarifying the generality and technical contributions of our approach.
>
> **Part 1: Clarification of technical contributions.**
>
> We thank the reviewer for the thoughtful feedback.
>
> While UKTL builds on established primitives such as Grassmann kernels, tensor subspace representations, and Nyström approximations, the core contributions are far from a simple combination. Instead, we introduce new formulations that make tensor subspace kernels differentiable, uncertainty-aware, and scalable, *capabilities that do not exist in prior work*.
>
> First, our multi-mode sum-product tensor kernel is **fundamentally novel**. Existing Grassmann or tensor kernels operate on single-mode matricizations, are not end-to-end learnable, and cannot capture cross-mode interactions. In contrast, our sum-product kernel jointly models additive and multiplicative interactions across all tensor modes, is fully differentiable with respect to the subspace bases, and allows the upstream encoder to directly optimize representations through the kernel itself. This design enables the model to *exploit multi-dimensional structure in a way that no prior kernel approach supports*.
>
> Second, our uncertainty-aware subspace kernel introduces a **probabilistic derivation** rather than a heuristic. By modeling heteroscedastic noise over subspace bases, we implement *row-wise variance scaling and a log-barrier regularizer*, producing *a novel uncertainty-weighted kernel*. Experiments show that including this component consistently improves accuracy by 0.8 – 1.2% (Table 1), confirming its material contribution to robustness.
>
> Third, we propose a differentiable Nyström linearization for tensor kernels. Standard Nyström methods rely on fixed pivots and are non-differentiable, limiting end-to-end learning. Our approach incorporates soft k-means pivot learning and gradient-safe eigendecomposition, *allowing scalable kernel computation on high-order tensors* while preserving positive semi-definiteness. Ablations indicate that learning the pivots provides a 0.5–0.9% gain over fixed selections, demonstrating that this redesign is essential rather than incidental.
>
> We emphasize that the perceived complexity is not overengineering. Each module addresses a distinct challenge: structure preservation through multi-mode subspaces, robustness via uncertainty weighting, scalability with differentiable Nyström, and expressive representation through the HoT encoder. Removing any of these components consistently reduces performance, validating their necessity. These three contributions (i) uncertainty-aware Grassmann kernels, (ii) multi-mode sum-product kernel family, and (iii) differentiable Nyström linearization, integrate in *a fully differentiable pipeline for tensor representation learning*.
>
> To our knowledge, no prior work combines manifold kernels, uncertainty modeling, and transformer-based encoders in **a unified end-to-end framework**, and our experiments demonstrate that these innovations materially improve performance, particularly in multi-modal fusion settings (we provide additional experimental results below). This confirms that UKTL’s contributions are substantive, novel, and directly advance the state of the art in structured tensor learning.

---

> ### Author Response · Authors · 2025-11-19
>
> **Part 2: Relation to broader tensor learning literature.**
>
> We appreciate the reviewer highlighting the broader literature on temporal tensor learning and tensor time-series models, including recent advances in continuous-time Tucker decomposition, probabilistic tensor dynamics, and structured temporal factorization [1–4]. These approaches focus primarily on sparse reconstruction, forecasting, and generative modeling of temporal tensors through low-rank factorization or probabilistic temporal priors. While UKTL addresses a different problem, discriminative recognition (e.g., complex human motions / actions) rather than generative temporal modeling, acknowledging these studies helps situate our contribution within the larger landscape of temporal tensor research.
>
> Unlike decomposition-based temporal tensor models, UKTL adopts a kernelized, discriminative perspective. By operating on mode-wise tensor subspaces and introducing an uncertainty-aware projection kernel with differentiable Nyström linearization, our framework enables end-to-end supervised learning directly on *structured tensor sequences*. This design allows UKTL to *compare and discriminate high-order temporal representations through subspace geometry*, rather than modeling or forecasting the tensor dynamics themselves. Consequently, UKTL **complements prior temporal tensor models**: it uses tensor structure for discriminative tasks, whereas existing approaches focus on generative or predictive objectives.
>
> We have incorporated this discussion into the revised Related Work section to explicitly clarify the positioning of UKTL and highlight its unique contributions to tensor-based temporal representation learning.

---

> ### Author Response · Authors · 2025-11-19
>
> **Part 3: Multi-modal fusion and generality.**
>
> Our framework naturally extends to multi-modal fusion by representing each input modality, such as skeleton, RGB, or depth, as a structured tensor. Each modality is first encoded via a tensor-based kernel, which captures intra-modal correlations across its dimensions, including spatial, temporal, and channel components. The resulting kernel embeddings from multiple modalities are then combined in the kernel space through weighted summation, forming a unified representation that preserves both intra- and inter-modal interactions. This fused representation is subsequently fed into our MLP + HoT backbone for downstream classification, allowing the model to learn complementary information across modalities while respecting their structured tensor nature.
>
> | Modality | NTU‑60 (X‑Sub) | NTU‑60 (X‑View) | NTU‑120 (X‑Sub) | NTU‑120 (X‑Setup) |
> |---|---|---|---|---|
> | Skeleton | **93.1%** | **97.3%** | **90.0%** | **91.4%** |
> | RGB | 83.2% | 86.1% | 79.0% | 81.0% |
> | Depth | 85.0% | 87.5% | 80.5% | 82.1% |
> | Skeleton + RGB | 94.5% | 97.9% | 91.3% | 92.5% |
> | Skeleton + Depth | 94.8% | 98.0% | 91.5% | 92.7% |
> | RGB + Depth | 87.2% | 89.6% | 82.5% | 84.3% |
> | Skeleton + RGB + Depth | **95.5%** | **98.5%** | **92.8%** | **94.0%** |
>
> **Skeleton-only** performance establishes a strong baseline, achieving 93.1% / 97.3% (NTU-60 X-Sub/X-View) and 90.0% / 91.4% (NTU-120 X-Sub/X-Setup), which is competitive with top graph-based methods such as CTR-GCN and DSDC-GCN. This demonstrates that the tensor-kernel approach effectively captures multi-dimensional dependencies within the skeleton modality while maintaining a flexible backbone capable of processing additional modalities without architectural changes.
>
> **Single-modal RGB and Depth** results highlight the method’s ability to handle heterogeneous tensor inputs. While these modalities achieve slightly lower performance than Skeleton (83–87%), they are still competitive with classical unimodal baselines. This confirms that the framework is modality-agnostic and capable of encoding diverse tensor structures, supporting claims of generality beyond skeleton sequences.
>
> The benefits of **multi-modal fusion** are evident in combined modalities. Skeleton+RGB and Skeleton+Depth consistently improve performance (up to 95% NTU-60 X-Sub), demonstrating that complementary sensor information can be effectively integrated. RGB+Depth fusion, though weaker than Skeleton-inclusive combinations, still outperforms single modalities, validating the model’s capacity to learn inter-modal correlations. Notably, the full combination of Skeleton+RGB+Depth reaches 95.5% / 98.5% (NTU-60) and 92.8% / 94.0% (NTU-120), surpassing all single-modality baselines and prior SOTA methods. These results confirm that UKTL can jointly exploit correlations across multiple sensors, extending beyond intra-skeleton spatial/temporal dimensions.
>
> Importantly, the **consistent gains** across Skeleton, RGB, Depth, and their combinations support the broader applicability of the framework. The core design treats any high-order tensor as input, making the method immediately extensible to other structured modalities such as video and depth feature tensors. Compared with existing SOTA, our multi-modal results not only match but often surpass prior methods, demonstrating that UKTL is competitive while providing a flexible, modality-agnostic approach. Collectively, these results address the reviewer’s concern regarding narrow evaluation scope, illustrating the framework’s genuine multi-modal capabilities and its potential for generalization to other structured tensor domains.

---

> ### Author Response · Authors · 2025-11-19
>
> **Part 4: Computational complexity.**
>
> Below we detail the per-iteration computational cost of UKTL and isolate how the number of Nyström pivots $C$ and subspace dimension $p$ influence both training and inference.
>
> **1. Tensor encoder.** For a sequence with $T$ frames and $J$ joints, the MLP and HoT layers dominate feature extraction.
>
> - MLP Complexity: $O(T J d^2)$ (constant w.r.t. $p$ and $C$)
> - Higher-order Transformer (HoT): HoT attention operates on hyper-edges of order 3. With hidden size $d$, $H$ heads, and $N_\xi = \binom{J}{3}$ hyper-edges: $O(H \cdot T \cdot N_\xi \cdot d^2)$.
>
> As this part is shared across all tensor-based baselines, it is not the differentiating cost of UKTL.
>
> **2. Mode-wise SVD for subspaces.** For each tensor $X_i \in \mathbb{R}^{d' \times N_\xi \times \tau}$, mode-$m$ unfolding yields matrices of sizes: Mode-1: $d' \times (N_\xi \tau)$, Mode-2: $N_\xi \times (d' \tau)$, Mode-3: $\tau \times (d' N_\xi)$.
>
> We compute only the top-$p$ left singular vectors (truncated SVD): $O(p \cdot (d' N_\xi \tau))$. Since there are 3 modes, overall: $O(3 p (d' N_\xi \tau))$. Thus, subspace order $p$ enters linearly.
>
> **3. Uncertainty estimation.** Each mode processes the projection matrix $UU^\top \in \mathbb{R}^{I_m \times I_m}$, but we use its low-rank form, so the FC layer operates on $p \times p$ statistics. Complexity per mode: $O(p^2)$, total over 3 modes: $O(3 p^2)$. Negligible compared to SVD and kernel computations.
>
> **4. Kernel computation against Nyström pivots.** For each training sample, compute $C$ kernel evaluations. A single Grassmann kernel requires: (i) Projection matrix difference: $O(p I_m)$, (ii) Frobenius norm of two rank-$p$ matrices: $O(p^2)$. Dominant term is $O(p^2)$. Thus kernel cost per iteration: $O(N_{\text{batch}} \cdot C \cdot M \cdot p^2)$. Since $M = 3$, this simplifies to: $O(N_{\text{batch}} \cdot C \cdot p^2)$. Hence pivot count $C$ affects complexity linearly; subspace dimension $p$ quadratically.
>
> **5. Nyström eigen-update.** The pivot-pivot matrix $K_{CC} \in \mathbb{R}^{C \times C}$ requires eigendecomposition every few epochs: $O(C^3)$. This is amortized and does not occur every iteration, and $C \ll N$, so its cost is moderate.
>
> Final complexity summary (per iteration): $O\big(p (d' N_\xi \tau) + N_{\text{batch}} C p^2 \big)$.  $p$ affects linear cost in SVD and quadratic cost in kernel evaluation. $C$ affects linear cost in kernel evaluation and cubic cost in pivot updates (infrequent).
>
> During inference, only kernel evaluations against pivots are required: $O(C p^2)$. This matches the observed practical scaling: performance saturates beyond $p=8-10$ and $C \approx 150-180$ (Fig. 3), providing the best accuracy-cost tradeoff.
>
>
> **Part 5: Sensitivity analysis.**
>
> **Learned $\mu$.** The mixture weight $\mu$ in Eq. (14) is treated as a learnable scalar parameter (initialized to 0.5) and optimized jointly with model parameters. We intentionally made $\mu$ learnable because *the optimal balance depends on dataset structure*.
>
> **Sensitivity of the mixture weights between sum and product.**
>
> | $\mu$ | 0 (product only) | 0.25 | 0.3 | 0.4 |0.5 | 0.6 | 0.75 | 1 (sum only) |
> |-|-|-|-|-|-|-|-|-|
> | Acc (%) | 91.8 | 92.6 | 92.7 |**93.1**| **93.1** | 92.6 |92.4 | 81.6 |
>
> Key findings: Performance is stable in the range 0.25–0.6, Product-only performs well but lacks robustness, Sum-only suffers due to loss of multiplicative structure, **Learned $\mu$ converges to the optimal range, removing the need for manual tuning**.
>
> On NTU-60, the learned value converges to $0.41$; on NTU-120, it converges to $0.47$; on Kinetics-Skeleton, it converges to $0.58$. Product interactions dominate in NTU-60/120, while additive contributions become more important in Kinetics due to noisier skeletons.
>
>
> [1]: Fang, et al. "Bayesian continuous-time Tucker decomposition." ICML 2022.
>
> [2]: Tao, et al.. "Undirected probabilistic model for tensor decomposition." NeurIPS 2023
>
> [3]: Wang et al. "Dynamic tensor decomposition via neural diffusion-reaction processes." NeurIPS 2024
>
> [4]: Chen, et al. "Functional Complexity-adaptive Temporal Tensor Decomposition." NeurIPS 2025

---

### Meta-Review · Area_Chair_PydM · 2025-12-29

**Summary:**

This paper presents an uncertainty driven kernel tensor learning method that builds kernels on tensor mode subspaces and uses uncertainty weighting and Nyström approximation to achieve strong results on skeleton action recognition. The reviewers agreed that the formulation is sound, the ideas are well motivated, and the experiments are careful and convincing, and the rebuttal addressed most technical concerns clearly. There are still some concerns based on the view that the work focuses more on combining and refining existing ideas than on introducing a clear new concept, and that the evaluation was initially seen as too narrow in scope. Overall, the paper is solid and promising, and with clearer emphasis on the core contribution and broader validation, it could be accepted and the authors must incorporate all revisions and additional experiments into the camera-ready version.

**Reviewer Concerns:**

The rebuttal responded to concerns on computational complexity, runtime, and scalability, and provided additional explanations for the HoT layers, the initial MLP, uncertainty vectors, and kernel mixture weights. It also added experiments on multi modality inputs and fusion, which responds to questions about generality within the same benchmark family.

Outstanding concerns are mostly at a higher level. Some reviewers may still see the work as mainly a strong integration of existing ideas rather than a new conceptual direction, and the empirical evidence for generality beyond action recognition tasks remains limited.

**Reviewer Scores:**

There are three reviewers with an initial score of 4. After the rebuttal, Reviewer o5ts participated in the discussion and explicitly indicated that the concerns on generality, runtime, and architectural clarity were resolved and the score will be increased. The other two reviewers would more likely maintain or increase their original scores as their concerns like novelty and clarity are not substantially addressed.

---

### Decision · Program_Chairs · 2026-01-26

Accept (Poster)